# CONSTRAINED ADAPTIVE REJECTION SAMPLING

## ABSTRACT

Language Models (LMs) are increasingly used in applications where generated outputs must satisfy strict semantic or syntactic constraints. Existing approaches to constrained generation fall along a spectrum: greedy constrained decoding methods enforce validity during decoding but distort the LM's distribution, while rejection sampling (RS) preserves fidelity but wastes computation by discarding invalid outputs. Both extremes are problematic in domains such as program fuzzing, where both *validity* and *diversity* of samples are essential. We present *Constrained Adaptive Rejection Sampling* (CARS), an approach that strictly improves the sample-efficiency of RS without distributional distortion. CARS begins with unconstrained LM sampling and adaptively rules out constraint-violating continuations by recording them in a trie and subtracting their probability mass from future draws. This adaptive pruning ensures that prefixes proven invalid are never revisited, acceptance rates improve monotonically, and the resulting samples exactly follow the constrained distribution. In experiments on a variety of domains—e.g., program fuzzing and molecular generation—CARS consistently achieves higher efficiency—measured in the number of LM forward passes per valid sample—while also producing stronger sample diversity than both Greedy Constrained Decoding (GCD) and methods that approximate the LM's distribution.

## 1 INTRODUCTION

Many applications of Language Models (LMs) require outputs that are not just fluent, but also satisfy strict structural or semantic constraints (Geng et al., 2025). Examples include ensuring syntactic validity in programming languages, adherence to schemas in data formats, or generating programs in restricted fragments of a given language.

This issue has motivated extensive work on *constrained generation*, i.e., methods for sampling using a language model so that its outputs satisfy a given structural or semantic specification. Two fundamental requirements emerge in this problem space:

- **Fidelity:** do samples follow the *exact* LM distribution conditioned on the constraint, or only an approximation?
- **Efficiency:** how many LM forward passes are required to obtain valid samples?

Most existing methods, which fall into three families, succeed on one axis but sacrifice the other.

**Exact methods.** Rejection Sampling (RS) is the canonical example. It produces unbiased samples from the true constrained distribution but wastes computation by discarding the overwhelming majority of candidates (e.g., $< 1\%$ acceptance in many structured domains).

**Static approximation methods.** Greedy constrained decoding (GCD) enforces validity by masking tokens that lead to constraint failure during generation (Geng et al., 2023; Park et al., 2025). While efficient, GCD provably distorts the conditional distribution (Tam et al., 2024; Park et al., 2024), often degrading downstream performance (Tam et al., 2024). It can even fail to terminate in some cases (e.g., repeatedly "opening brackets" without producing a complete valid sequence).

**Asymptotic approximation methods.** These methods include iterative over-approximations of invalid prefixes (Park et al., 2024), Monte Carlo and sequential Monte Carlo approaches that resample from inexact constrained distributions to approximate the desired distribution (Anaya Gonzalez et al., 2025), and other MCMC-style refinements (Lew et al., 2023; Melcer et al., 2024b). All

of these techniques are guaranteed to converge to the correct distribution in the limit, but they provide no principled stopping rule: early samples can be arbitrarily biased, and efficiency depends heavily on how many candidates must be drawn before the approximation stabilizes. Moreover, these methods require hyperparameter tuning (e.g., number of MCMC steps $k$ or SMC particles $M$) without providing practitioners with a principled way to determine if these hyperparameters yield sufficient distributional accuracy.

Thus, the current landscape reflects a fundamental tradeoff: *exactness without efficiency, or efficiency without exactness*. This tradeoff becomes especially limiting in domains where performance depends on generating sets of diverse, constraint-satisfying outputs from the same LM context—such as program fuzzing (Anaya Gonzalez et al., 2025) or molecule discovery (Wang et al., 2023). In these cases, the key desideratum is not only fidelity, but also **amortized efficiency**: across many samples, can the average number of LM forward passes per valid output be kept low? Existing asymptotic methods achieve amortized efficiency only asymptotically, and only at the cost of biased early samples. *What is missing is an exact algorithm that is amortized-efficient in practice.*

We propose *Constrained Adaptive Rejection Sampling* (CARS), an exact method that combines the fidelity of RS with the efficiency benefits of constraint-aware decoding. CARS builds on Adaptive Rejection Sampling (ARS) (Mansinghka et al., 2009), which adaptively avoids repeating rejected samples. CARS goes further: as each LM sample is generated, the algorithm uses constrained decoding algorithms to identify not only the rejected output but also all nearby continuations of its partial prefixes that would inevitably violate the constraint. Each invalid prefix is recorded in a trie, and its probability mass is subtracted from future generations, ensuring monotonic improvements in acceptance rate while preserving the exact constrained distribution.

Although, in theory, CARS could still require many rejections for adversarial constraints, we argue—and demonstrate empirically—that real-world constrained LM tasks fit the CARS setting well: most constraints are prefix-checkable (e.g., validity according to a context-free grammar or type system), inexpensive to enforce, and highly informative for pruning. This makes CARS asymptotically efficient in practice while remaining exact, thus setting a new state-of-the-art for sampling from the exact LM's distribution in the presence of constraints.

We make the following contributions. We introduce **CARS**, a new algorithm for constrained LM generation that achieves exactness with practical efficiency by leveraging constraint structure (Section 3). Our evaluation shows that CARS achieves higher acceptance rates, stronger diversity, and lower amortized cost than existing constrained sampling methods (Section 4).

## 2  EXACT CONSTRAINED SAMPLING

In this section, we formalize the problem of sampling from a language model (LM) conditioned on a constraint (i.e., constrained sampling), define our key desiderata of a good constrained sampling algorithm, and describe how existing constrained sampling algorithms do not meet such desiderata. We follow the definitions proposed by Park et al. (2024) and Anaya Gonzalez et al. (2025).

Let $\Sigma_\$$ be a set of tokens including an end-of-sequence marker \$, and let $\Sigma = \Sigma_\$ \setminus \{\$\}$. We consider sequences from the set $\Sigma^* \$^?$ (i.e., sequences of tokens that may have \$ only at the end). We write $u \preceq w$ to denote that a sequence $u$ is a prefix of a sequence $w$. For a set of sequences $\mathcal{L}$ we write $\text{prefix}(\mathcal{L})$ to denote the set of prefixes of sequences in $\mathcal{L}$—i.e., $\text{prefix}(\mathcal{L}) = \{u \mid \exists w \in \mathcal{L}.\, u \preceq w\}$—and $\text{ext}(\mathcal{L})$ to denote sequences extending a sequence from $\mathcal{L}$—i.e., $\text{ext}(\mathcal{L}) = \{w \in \Sigma^* \$^? \mid \exists u \in \mathcal{L}.\, u \preceq w\}$.

**Language Models.**  An (autoregressive) language model is given by next-token conditional probability distributions of the form $P(ua \mid u)$, where $u \in \Sigma^*$ and $a \in \Sigma_\$$—denoting the probability that a sequence $u$ is followed by a token $a$. This definition extends to longer continuations: $P(ua_1 \ldots a_n \mid u) = \Pi_{i=1}^n P(ua_1 \ldots a_i \mid ua_1 \ldots a_{i-1})$.

More generally, for any prefix $u \in \Sigma^*$ and suffix $w \in \Sigma^* \$^?$, we write $P(w \mid u)$ for the probability that the model generates $w$ as a continuation of $u$ before either producing the end-of-sequence symbol \$ or reaching length $|w|$. We also write $P(w \mid u) = 0$ when $u$ is not a prefix of $w$. For technical reasons, we assume that $\sum_{w \in \Sigma^* \$} P(w) = 1$, which means that almost surely the \$ marker will be produced at some moment (the probability that an infinite word without any \$ marker will be

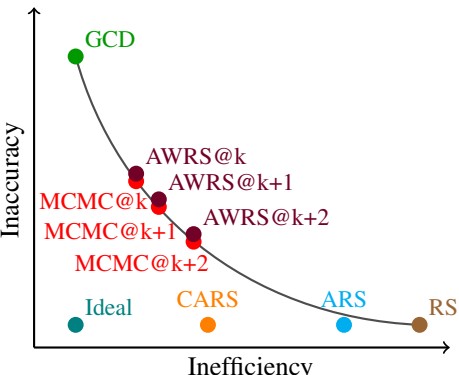

Figure 1: Trade-off between inaccuracy and inefficiency across sampling methods.

produced is 0; this can be achieved by modifying the LM to generate $ with probability 1 after a maximum sequence length).

**Constraints.** Given a language model $P$ and a constraint, the goal of constrained sampling is to sample sequences that satisfy the constraint. Formally, a constraint is just a set $\mathcal{L} \subseteq \Sigma^*\$$ of sequences (satisfying the constraint). In practice, the set $\mathcal{L}$ may be given in many possible ways, e.g., as a regular language, context-free grammar (CFG), or some logical condition.

While some constraints are computationally expensive to verify, in this work and in our experiments, we focus on constraints that can be incrementally evaluated over the entire token vocabulary. This means that we have a fast algorithm that given a word prefix $u$ generates a vector of answers, saying for each possible next token $a \in \Sigma_\$$ whether $ua \in \text{prefix}(\mathcal{L})$ (i.e., whether $ua$ can be continued into a full sequence satisfying the constraint). In particular, this holds for context-free grammars (CFGs) (AI, 2025; Park et al., 2025), which can, for instance, describe the set of syntactically valid programs in a programming language or enforce the correct structure of a JSON object.

An example domain where constrained decoding is used to generate many diverse samples is asking a language model to generate SQLite regression test files that exercise as many distinct execution paths in the SQLite engine as possible (see Section 4.1). To target specific components of the database, each file must satisfy the syntactic and semantic rules of the SQLite test-script grammar.

**Exact Constraint-Aligned Sampling.** Constraint-aligned sampling aims to generate sequences from a model $P$ that satisfy a given set of hard constraints, while preserving the model's underlying distribution. Formally, this corresponds to sampling sequences from the constrain set $\mathcal{L}$, where the probability of each word $w \in \mathcal{L}$ should be $P^{\mathcal{L}}(w) = \frac{P(w)}{\sum_{w' \in \mathcal{L}} P(w')}$.

In this work, we focus on designing an algorithm that samples **exactly from the conditional distribution** $P^{\mathcal{L}}$ while being **more efficient than existing exact methods**.

**Existing Exact Methods.** Rejection Sampling (RS) repeatedly draws outputs from the LM and discards those violating the constraint. However, RS is highly inefficient when valid sequences are rare under the LM, which is common in structured domains. Adaptive Rejection Sampling (ARS) (Gilks et al., 2018) improves upon RS by dynamically avoiding previously observed invalid samples. It remains exact but only adapts to prefixes or outputs that have been explicitly seen to fail.

Exploiting additional prefixes is the key distinguishing factor that makes Constrained Adaptive Rejection Sampling (CARS) more efficient. For instance, in our fuzzing benchmarks we observe cases in which ARS maintains a rejection rate higher than 99% even after 1,000 samples, whereas CARS lowers rejection to rates in the 70-95% range after just 100 samples (Section 4.1).

---

**Algorithm 1:** CARS algorithm

---

**Input:** Constraint language $\mathcal{L} \subseteq \Sigma^*\$$
**Output:** Infinite sequence of samples drawn from the constrained distribution $P^{\mathcal{L}}$

1 $\mathcal{W} \leftarrow \emptyset$ ;                          // initialize invalid prefixes
2 **while** *true* **do**
    // $R^{\mathcal{W}}$ is adaptively reshaped to avoid invalid samples in $\mathcal{W}$
3   $w \sim R^{\mathcal{W}}$ ;     // sample from adaptively reweighted distribution
4   **if** $w \in \mathcal{L}$ **then**
5      **yield** $w$ ;                                // yield a sample
6   $\mathcal{W} \leftarrow \mathcal{W} \cup \text{INVALID}(w, \mathcal{L})$ ;     // add new invalid prefixes from $w$

---

## 3 CONSTRAINED ADAPTIVE REJECTION SAMPLING

The Constrained Adaptive Rejection Sampling (CARS) algorithm maintains a set $\mathcal{W}$ to rule out invalid prefixes that have been discovered during sampling, and uses the probability of such prefixes according to the LM to compute an adaptively reshaped version $R^{\mathcal{W}}$ of the sampling distribution that is such that future sampling iterations will provably not repeat past mistakes. This lets us retain the exact distributional fidelity of rejection sampling while avoiding wasted computation on already-eliminated sequences.

Figure 1 clarifies the goal of this paper: among all the exact methods, CARS is the most efficient.

**Example 1** (Arithmetic Expressions). *As a running example, consider a toy grammar for arithmetic expressions over digits:*

$$E \quad ::= \quad d\$ \mid d + E \quad where \; d \in \{0, 1\}.$$

*Here, strings like `1+0+1` satisfy the constraint—i.e., they are accepted by the grammar—while `0++` or `+1` are not. We will use this grammar to illustrate how $\mathcal{W}$, $R^{\mathcal{W}}$, and the update step evolve during sampling.*

The rest of this section explains the pieces of Algorithm 1: how $R^{\mathcal{W}}$ is defined and sampled, how the prefix set $\mathcal{W}$ is maintained, and how different update strategies for $\mathcal{W}$ provide different benefits.

**Tracking Invalid Prefixes.**   CARS maintains a finite set $\mathcal{W} \subseteq \Sigma^*\$^?$, called *invalid prefixes*. By construction, $\mathcal{W}$ is disjoint from $\text{prefix}(\mathcal{L})$, the set of valid prefixes. For each sequence $u \in \Sigma^*\$^?$, the algorithm implicitly tracks a value $p_u$ representing the probability of extending $u$ into a complete sequence that avoids $\mathcal{W}$:

$$p_u = \sum_{w \in \Sigma^*\$ \setminus \text{ext}(\mathcal{W})} P(w \mid u).$$

The values $p_u$ are updated whenever $\mathcal{W}$ is updated, and are jointly represented with $\mathcal{W}$ in the same trie data structure.

These values satisfy the following equation for words without the end-of-sequence marker $\$$:

$$\forall u \in \Sigma^* \qquad p_u = \sum_{a \in \Sigma_\$} P(ua \mid u) \cdot p_{ua}. \tag{1}$$

We additionally observe that

$$p_u = 0 \qquad \text{if } u \text{ starts with a known invalid prefix, i.e., } u \in \text{ext}(\mathcal{W}), \text{ and}$$
$$p_u = 1 \qquad \text{if } u \text{ cannot be extended to any known invalid prefix, i.e., } u \notin \text{prefix}(\mathcal{W}).$$

In the arithmetic-expression grammar from Example 1, we may at some moment discover that `0++` is invalid. Then Line 6 adds this prefix to $\mathcal{W}$, and thus any string $u$ that has prefix `0++` has $p_u = 0$.

Initially, we have $\mathcal{W} = \emptyset$, and hence $p_u = 1$ for all sequences $u$—i.e., we have not yet proven that any sequence can violate the constraint and thus their probability of extending to a constraint-satisfying sequence is still upper-bounded by 1.

**The Distribution $R^{\mathcal{W}}$.**    At any iteration, given the current set $\mathcal{W}$, CARS samples from a reweighted distribution $R^{\mathcal{W}}$ over the set of sequences $\Sigma^* \setminus \text{ext}(\mathcal{W})$ that so far has not been proven incorrect. It is convenient to represent the probabilities associated to each prefix in $\text{prefix}(\mathcal{W})$ using a trie structure. Elements of $\mathcal{W}$ are leaves of the trie, and for internal nodes we store the actual values of $p_u$ calculated according to Equation (1). Note that we do not need to store any more values of $p_u$, as for other sequences we have that either $p_u = 0$ or $p_u = 1$.

When a new sequence $w$ is added to $\mathcal{W}$, we add the corresponding leaf to the trie and set its probability $p_w$ to 0. This update is then propagated upward in the trie: whenever a child probability $p_{ua}$ decreases by $x$, the parent $p_u$ decreases by $P(ua \mid u) \cdot x$.

For example, suppose 0++ is added to $\mathcal{W}$. The trie node corresponding to 0++ becomes a leaf ($p_{0++} = 0$). Then $p_{0+}$ is decreased proportionally to the probability of extending 0+ with another +, thus subtracting the probability of entering this invalid path (which the trie will now disallow).

For a given set $\mathcal{W}$, the quantity $p_\varepsilon = \sum_{w' \in \Sigma^*\$\setminus\text{ext}(\mathcal{W})} P(w')$ determines the total probability of all sequences avoiding $\mathcal{W}$; we can then define the distribution $R^{\mathcal{W}}$ on sequences $w \in \Sigma^*\$ \setminus \text{ext}(\mathcal{W})$ to be $R^{\mathcal{W}}(w) = \frac{P(w)}{p_\varepsilon}$. The probabilities sum to 1. Importantly, we can sample from $R^{\mathcal{W}}$ left-to-right: for $u \in \Sigma^*$ and $a \in \Sigma_\$$, $R^{\mathcal{W}}(ua \mid u) = P(ua \mid u) \cdot \frac{p_{ua}}{p_u}$.

In our arithmetic-expression grammar, once 0++ is ruled out, whenever the prefix 0+ is visited, the probability of sampling another + vanishes, and the model is effectively forced to choose some token other than + instead.

Because $\mathcal{W}$ is finite, so is $\text{prefix}(\mathcal{W})$. When we sample a sequence prefix $u$ that does not belong to $\text{prefix}(\mathcal{W})$, we have $p_{ua} = p_u = 1$ (and so on for any extension of $ua$) and $R^{\mathcal{W}}$ reduces to the original distribution $P$. Thus, sampling from $R^{\mathcal{W}}$ almost surely terminates with the $ token.

**Updating $\mathcal{W}$.**    The update step $\mathcal{W} \leftarrow \mathcal{W} \cup \text{INVALID}(w, \mathcal{L})$ at Line 6 determines the efficiency of CARS: adding more information reduces the sample-rejection rate in future iterations. Any strategy for updating $\mathcal{W}$ is valid provided that only prefixes outside of $\text{prefix}(\mathcal{L})$ are added to $\mathcal{W}$. Existing rejection sampling approaches can be framed as update strategies:

*Rejection Sampling (RS):* never updates $\mathcal{W}$, simply retries until success.

*Adaptive Rejection Sampling (ARS):* adds only the rejected string $w$ or its shortest invalid prefix to $\mathcal{W}$. In the arithmetic-expression grammar, if the sampler produces 0+++, then ARS only adds the shortest invalid prefix 0++ to $\mathcal{W}$ (Figure 2).

*Rejection Sampling with constrained First Token (RSFT):* a variant of ARS that limits invalid prefixes to length 1. When a sequence is rejected, RSFT only adds single-token invalid prefixes to $\mathcal{W}$, ensuring future samples never start with an invalid token while allowing subsequent tokens to be sampled freely. We adopt this method as a baseline in our evaluation to assess how much of the probability mass is "wasted" on sequences with invalid starting tokens. In the arithmetic-expression grammar, regardless of the produced sample, RSFT adds the prefixes +, 2, etc. to $\mathcal{W}$, preventing any sequence from *starting* with an invalid token.

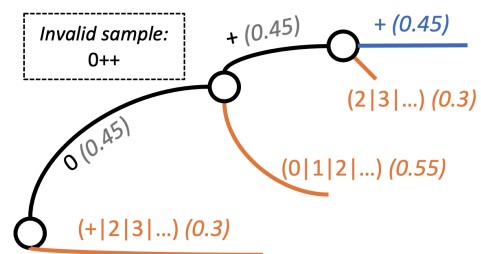

Figure 2: Invalid sample 0++ for the arithmetic grammar in Example 1. The sequence ending in the blue token is invalid for both ARS and CARS, whereas the sequences ending with orange tokens are only considered invalid by CARS. With the example probabilities in parenthesis, ARS reduces the future rejection probability by $0.09 \approx 0.45 * 0.45 * 0.45$ whereas CARS reduces it by $0.63 \approx 0.3 + 0.45 * 0.55 + 0.45 * 0.45 * 0.45$.

*Constrained Adaptive Rejection Sampling (CARS):* the update strategy that *(i)* adds to $\mathcal{W}$ the shortest prefix $u$ of $w$ that is not in $\text{prefix}(\mathcal{L})$, and *(ii)* for every proper prefix $u$ of $w$ and for every token $a$ such that $ua \notin \text{prefix}(\mathcal{L})$, adds $ua$ to $\mathcal{W}$. In the arithmetic-expression grammar, if the LM produces 0++, then CARS adds the shortest invalid prefix 0++ to $\mathcal{W}$, but also all invalid continuations of its shorter prefixes—e.g., +, 2, 3, . . . (invalid continuations of the empty prefix), 10, . . . (invalid

continuations of the prefix `1`), and `0+a`, `0++`, … (invalid continuations of the prefix `1+`). Point (ii) applies even if the LM produces a valid sample, e.g., while producing a valid sequence `0+1$`, the same prefixes and their invalid continuations are added to $\mathcal{W}$.

**CARS is exact.** CARS produces unbiased samples from the true constrained distribution and the sample-acceptance rate increases monotonically. Rather than simply eliminating individual rejected sequences (as in ARS), CARS identifies and prunes entire families of invalid continuations at each prefix position. This structural exploitation means that convergence behavior depends on both the grammar's branching structure and the LM's token distribution over invalid regions.

**Theorem 1.** *The CARS algorithm samples an element of $\mathcal{L}$ according to the target distribution $P^{\mathcal{L}}$. Moreover, the adaptive updates performed in Line 6 of the algorithm monotonically increase the probability that some sequence is yielded in Line 5 at subsequent loop iterations.*

*Proof.* Whenever a sequence is produced by the algorithm in Line 5, it comes from the distribution $R^{\mathcal{W}}$, restricted to sequences in $\mathcal{L}$. But in $R^{\mathcal{W}}$ the probability of each sequence $w \in \mathcal{L}$ is proportional to $P(w)$, and the same also holds for $P^{\mathcal{L}}$. The probability that a fixed sequence $w \in \mathcal{L}$ is produced by the algorithm equals $R^{\mathcal{W}}(w) = \frac{P(w)}{p_\epsilon}$. While $P(w)$ is a constant probability, the number $p_\epsilon$ monotonically decreases whenever we add a new invalid prefix to $\mathcal{W}$, causing that $R^{\mathcal{W}}(w)$ increases. The probability that some sequence is produced is just a sum of $R^{\mathcal{W}}(w)$ over all sequences $w \in \mathcal{L}$, hence it increases as well. $\square$

CARS's trie operations account for only 0.3% of total runtime. Detailed profiling across all benchmarks is provided in Appendix E.

## 4 EVALUATION

In this section, we evaluate CARS in terms of efficiency and the quality of its samples compared to other constrained sampling methods. Because CARS samples exactly from the target grammar-constrained distribution $P^{\mathcal{L}}$, there is no convergence issue. Instead, our focus is on: (i) how efficiently each method produces valid sequences, and (ii) how closely approximate methods (e.g., GCD) match the exact distribution produced by CARS. We evaluate on tasks that require generating many diverse outputs, as this setting best showcases and evaluates amortized efficiency.

In Section 4.1, we demonstrate that seeds generated using CARS improve coverage in fuzzing tasks over approximate methods. Section 4.2 extends the evaluation to molecular synthesis, again highlighting efficiency and constraint satisfaction in domains where diversity is crucial. Section 4.3 evaluates text-to-SQL generation, demonstrating that distributional fidelity translates to improved downstream execution accuracy while maintaining sample efficiency.

Additional evaluations on task-focused domains (PDDL planning, SyGuS benchmarks) are provided in Sections I and J.

**Baselines.** We compare CARS against GCD (which is a static inexact approximation), existing exact algorithms discussed in Section 3 (Rejection Sampling (RS), Adaptive Rejection Sampling (ARS) (Mansinghka et al., 2009), Rejection Sampling with constrained First Token (RSFT)), and a state-of-the-art approximate algorithm (MCMC) (Anaya Gonzalez et al., 2025). We omit Adaptive Sampling with Approximate expected futures (ASAp) (Park et al., 2024) as Anaya Gonzalez et al. (2025) have shown MCMC outperforms ASAp across all benchmarks. For the benchmarks in Section 4.2, Section 4.3 and Section I, we additionally evaluate Adaptive Weighted Rejection Sampling with Sequential Monte Carlo (AWRS) (Lipkin et al., 2025). These benchmarks were considered in that work, and the implementation of AWRS directly works on them. The RSFT algorithm (which only learns how to avoid incorrect first tokens) is a special case of CARS that is also a contribution of our work. We select the best settings from the original papers, and choose $k = 10$ steps for MCMC and $M = 10$ particles for AWRS.

**Metrics.** Our key metric is sampling efficiency, measured as the number of LM generation calls—each generating a complete, but possibly invalid output sequence—needed to obtain a fixed number

of valid outputs for a given input. This metric captures the computational cost of each method and highlights how strategies such as CARS reduce wasted computation on ungrammatical sequences.

To evaluate the approximation effect of approximate sampling approaches, the ideal metric would be the distance between the empirical sample distribution and the target constrained distribution $P^{\mathcal{L}}$. However, computing this quantity exactly is impractical: the sequence space is often infinite, and $P^{\mathcal{L}}$ may be inaccessible for direct probability evaluation. We follow the approach by prior work (Park et al., 2024; Anaya Gonzalez et al., 2025) and use an approximate measure: the KL divergence between the empirical distribution of the generated samples $\tilde{P}^{\mathcal{L}}$ and the LM's distribution $P$. We obtain 100 samples for each sampling method for each task, and plot the mean KL divergence and 95% confidence interval ranges computed from bootstrapping across 3 different runs. Note that the empirical KL divergence can be greater than 0 even when we sample from the exact distribution.

## 4.1 GRAMMAR-BASED FUZZING

Anaya Gonzalez et al. (2025) demonstrated that constrained sampling can significantly improve seed generation for program fuzzers (Böhme et al., 2016; Herrera et al., 2021). Fuzzers randomly mutate an initial set of input program seeds to generate test cases that trigger different execution paths in a binary. By using grammars to prevent malformed inputs, Anaya Gonzalez et al. (2025) showed that the closer the LM's sampling aligns with the constrained distribution, the more execution paths the fuzzer can explore when generating additional inputs from these seeds.

**Benchmarks.** We evaluate on three targets with varying constraint complexity (details about benchmark choice in Section F.5): **JSON** processing (requires $\geq 3$ key-value pairs with a fixed first pair), **SQL** testing (mandates two do_test blocks per .test file), and **XML** parsing (requires 1 element declaration with $\geq 1$ ATTLIST in DOCTYPE). For each target, we consider two conditions: **prompts with grammar** and **prompts without grammar** (details in Section F.2).

**Metrics.** We evaluate *sample efficiency—the number of LM generations required to produce 100 valid samples, and *line coverage—the number of unique source code lines executed*, measured via LLVM instrumentation (llv, 2025). W use AFL++ (Fioraldi et al., 2020) as our fuzzer and run each fuzzing campaign for one hour. We impose a 2,000-sample cap on generation attempts.

**Findings.** Figure 3 reports results on the XML benchmark without grammar in the prompt.

RS and ARS fail to produce 100 valid samples within our 2,000-sample budget. CARS achieves the target with only 215 generations while RSFT requires 275—making our approaches the only practically feasible exact methods. This efficiency advantage holds across benchmarks: for JSON without grammar, CARS requires $\sim$130 generations versus $\sim$601 for RSFT (Table 7).

When comparing to inexact methods, CARS exhibits better KL divergence (Figures 3a and 5). Remarkably, at comparable sample complexity to MCMC (i.e., the vertical line in the plot), CARS shows significant improvements.

The improved faithfulness to the constrained distribution is also reflected in fuzzing line coverage: CARS-generated seeds achieve $\sim$8,815 lines covered (Figure 3b) compared to $\sim$7,115 for GCD and $\sim$8,765 for MCMC. Similar improvements are observed across other benchmarks (Section F.6), with CARS yielding a $\sim$12% improvement in coverage over GCD for JSON generation (both with and without grammar).

We note that SQL-with-grammar proved challenging: no exact approach produced 100 valid samples within budget for Qwen2.5-7B-Instruct, and only RSFT succeeded for Llama-3.1-8B-Instruct.

**Summary.** CARS is the only exact method that can handle some fuzzing benchmarks *efficiently* and its distributional fidelity translates to meaningful gains in downstream line coverage.

## 4.2 MOLECULAR SYNTHESIS

Constrained molecular generation is a central challenge in computational drug discovery and materials science (Kusner et al., 2017; Jin et al., 2019; 2020), where both *structural validity* and *chemical*

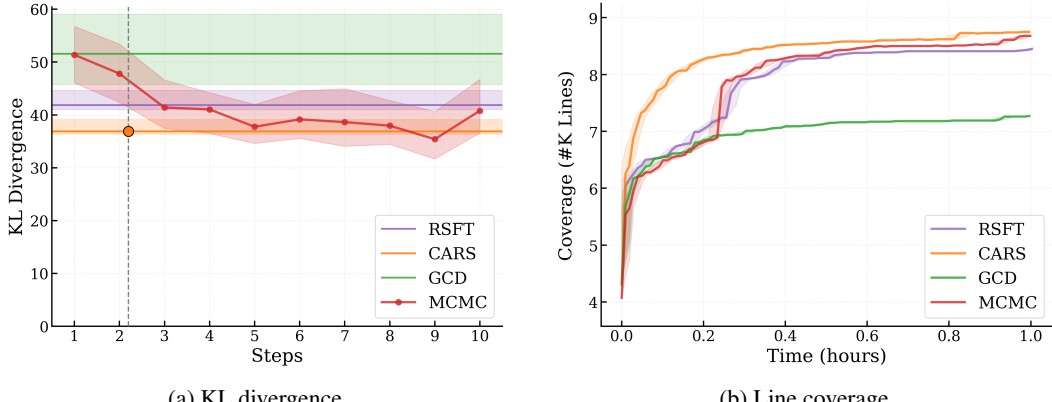

(a) KL divergence

(b) Line coverage

Figure 3: XML benchmark with grammar: (a) KL divergence for different sampling methods. (b) Line coverage achieved by fuzzing with generated seeds. Displayed KL for RSFT and CARS is non-zero (even though these methods are exact) because we compute an empirical estimate of KL. The vertical dashed line is the average number of steps MCMC would require to have the same sample efficiency as CARS (i.e., CARS averages 2.25 LM calls per sample.)

*diversity* across different samples are crucial for exploring chemical space effectively. This task requires producing valid SMILES (Weininger, 1988) strings that satisfy syntactic constraints (balanced parentheses, valid bonding) and semantic constraints (specific functional groups). We test whether CARS can improve sampling efficiency while maintaining high distributional fidelity in this setting.

**Benchmarks.** We evaluate on three structurally distinct molecular classes from prior work (Wang et al., 2023; Guo et al., 2022): Acrylates (32 example molecules), Chain Extenders (11 example molecules), and Isocyanates (11 example molecules). Each class requires both valid SMILES syntax and class-specific functional group constraints (e.g., acrylate `C=CC(=O)O` motifs). We adopt few-shot prompting with all available exemplars per class, and enforce both syntax and class-level constraints through grammars.

**Metrics.** We measure the four quality dimensions that are considered by Wang et al. (2023): (1) *Validity*: parseability via RDKit (RDKit); (2) *Diversity*: average pairwise Tanimoto distance over Morgan fingerprints (Rogers & Hahn, 2010); (3) *Retrosynthesis Score*: synthesizability via RetroStar (Chen et al., 2020); (4) *Membership*: correct classification in target class. For each method, we generate until obtaining 100 unique molecules in the grammar, excluding the example molecules provided in the prompt, subject to a 1000-sample cap. We averaged across three trials.

**Findings.** CARS consistently delivers advantages in both quality and efficiency. Table 1 shows the results. When accounting for standard deviation, CARS and the other exact methods all achieve the highest molecular diversity and validity. However CARS requires $4.3\times$ fewer samples than RS, and $1.3\times$ fewer samples than ARS. This reduction in wasted computation translates into substantial practical savings for molecular design pipelines.

The KL divergence of CARS is on average $1.5\times$ lower than MCMC and $1.8\times$ lower than AWRS. MCMC shows characteristic convergence behavior, starting with high divergence ($\sim$26) and gradually decreasing toward CARS's level over multiple steps, but never fully reaching the desired distributional accuracy. The differentiating factor is therefore efficiency, where CARS offers noticeable gains.

As expected, approximate methods suffer in most metrics, with AWRS showing particularly poor performance. Per-class breakdowns in Section G.5 confirm these trends across all molecular families and language models. We note that the *diversity* metric is a molecule-specific metric and is not the same as adherence to the exact probability distribution.

Table 1: Molecular generation performance across three chemical classes using Llama-3.1-8B-Instruct. Quality metrics show mean $\pm$ standard deviation over 3 trials. Sample efficiency shows samples required to generate 100 valid molecules. Bold indicates best performance.

| Method | Validity | Diversity | Retro Score | Membership | Samples/100 Valid |
|---|---|---|---|---|---|
| RS | $0.85 \pm 0.12$ | $0.83 \pm 0.06$ | $0.59 \pm 0.14$ | $0.82 \pm 0.12$ | $793 \pm 127$ |
| ARS | $\mathbf{0.87 \pm 0.09}$ | $0.83 \pm 0.07$ | $0.56 \pm 0.12$ | $\mathbf{0.85 \pm 0.10}$ | $220 \pm 34$ |
| RSFT | $0.82 \pm 0.15$ | $0.82 \pm 0.06$ | $0.53 \pm 0.11$ | $0.80 \pm 0.14$ | $765 \pm 89$ |
| CARS | $\mathbf{0.87 \pm 0.09}$ | $\mathbf{0.85 \pm 0.06}$ | $\mathbf{0.60 \pm 0.15}$ | $\mathbf{0.85 \pm 0.09}$ | $\mathbf{183 \pm 28}$ |
| GCD | $0.70 \pm 0.16$ | $0.84 \pm 0.05$ | $0.47 \pm 0.14$ | $0.72 \pm 0.13$ | $\mathbf{100 \pm 0}$ |
| AWRS | $0.02 \pm 0.02$ | $0.55 \pm 0.51$ | $0.00 \pm 0.01$ | $0.02 \pm 0.02$ | $1000 \pm 0$ |
| MCMC | $0.79 \pm 0.14$ | $0.84 \pm 0.03$ | $0.51 \pm 0.04$ | $0.77 \pm 0.10$ | $1000 \pm 0$ |

**Summary.** In molecular synthesis, where both validity and diversity are essential, CARS achieves the best of both worlds: unbiased sampling that preserves chemical diversity, together with large improvements in computational efficiency over standard rejection sampling.

## 4.3 TEXT-TO-SQL GENERATION

Constrained text-to-SQL generation is a fundamental task in natural language interfaces to databases (Zhong et al., 2017; Yu et al., 2019), and it requires producing SQL strings that satisfy both syntactic constraints (proper query structure, valid keywords) and semantic constraints (referencing valid table and column names from the schema). Unlike fuzzing and molecular synthesis where diversity is crucial, text-to-SQL focuses on solving concrete tasks: our goal is to assess whether exact samples from a constrained distribution are more likely to produce correct query executions (we also evaluate on similar task-focused domains in Sections I and J). We evaluate whether CARS can improve sampling efficiency while maintaining high execution accuracy in this setting.

**Benchmarks.** We evaluate on the development split of the Spider dataset (Yu et al., 2019), containing 1,034 examples across 200 databases with varying complexity. Each example consists of a natural language question paired with its corresponding database schema. We use Llama-3.1-8B-Instruct in a zero-shot setting (Section H confirms that CARS exhibit similar results for other models), and enforce SQL syntax through a context-free grammar specified in Lark.

**Metrics.** We measure *execution accuracy*, i.e., whether the generated SQL query produces the same results as the ground-truth query when executed on the test database. This metric captures both syntactic validity and semantic correctness. We report results averaged across four trials.

**Findings.** Exact methods achieve the highest execution accuracy, and CARS has superior sample efficiency among them. Table 2 shows that CARS attains 0.578 accuracy with only 1.11 samples per query on average, outperforming GCD's accuracy (0.525) by 5.3%. Among exact methods, CARS requires $2\times$ fewer samples than RS and $1.4\times$ fewer than ARS. Approximate methods show mixed results: MCMC and AWRS achieve similar accuracy to the exact methods, but require $9\times$ more samples than CARS. The KL divergence of CARS is on average $2.21\times$ lower than MCMC and $2.65\times$ lower than AWRS.

**Summary.** In text-to-SQL generation, CARS provides the best combination of accuracy and efficiency, achieving the highest execution accuracy while requiring fewer samples than both standard rejection sampling methods and approximate techniques that rely on hyperparameter tuning.

## 5 RELATED WORK

**Exact Methods** Our work is a direct improvement of ARS Gilks et al. (2018), which adaptively rejects samples that violate constraints. Our work also shares conceptual foundations with Tromble & Eisner (2006), who patch a language model with constraints as violations are discovered during decoding. However, their method targets argmax decoding over weighted FSAs, whereas we focus on sampling from arbitrary autoregressive LMs using dynamically generated prefix constraints.

Table 2: Text-to-SQL generation performance on Spider development set using Llama-3.1-8B-Instruct. Quality metrics show mean $\pm$ standard deviation over 4 trials. Bold indicates best performance.

| Method | Execution Accuracy | Total Samples | Samples/Query |
|--------|-------------------|---------------|---------------|
| RS | $0.576 \pm 0.014$ | $2126 \pm 155$ | 2.06 |
| ARS | $0.574 \pm 0.011$ | $1435 \pm 124$ | 1.39 |
| RSFT | $0.573 \pm 0.009$ | $1916 \pm 186$ | 1.86 |
| CARS | $\mathbf{0.578 \pm 0.013}$ | $1146 \pm 93$ | 1.11 |
| GCD | $0.525 \pm 0.011$ | $\mathbf{1034 \pm 0}$ | $\mathbf{1.00}$ |
| AWRS | $0.567 \pm 0.015$ | $10340 \pm 0$ | 10.00 |
| MCMC | $0.569 \pm 0.014$ | $10340 \pm 0$ | 10.00 |

**Static Approximation Methods.** Constrained decoding methods (Scholak et al., 2021; Beurer-Kellner et al., 2023; Geng et al., 2023; Melcer et al., 2024a) enforce constraints incrementally during token-by-token generation. When the constraint is a context-free grammar, this approach is often called Grammar-Constrained Decoding. While efficient at producing valid sequences, these methods modify the LM's probability distribution, resulting in biased samples. IterGen (Ugare et al., 2025) provides a programming framework with grammar-based navigation primitives for writing custom generation algorithms, but builds on Grammar-Constrained Decoding (via SynCode) and applies recurrence penalties during backtracking, further distorting the distribution. Gradient-based constrained decoding (Amini et al., 2024; Kumar et al., 2022) similarly steers generation toward satisfying soft or semantic constraints, but cannot guarantee validity (or faithfulness to the distribution) and is computationally expensive.

**Asymptotic Approximation Methods.** Adaptive Sampling with Approximate expected futures (ASAp) (Park et al., 2024) approximate grammar-aligned sampling by building an iterative overapproximation of the probability mass associated with invalid prefixes identified from previous samples. While in the limit this approach reaches the desired distribution, it does not do so *monotonically*—i.e., it can produce intermediate approximations that are very skewed.

Monte Carlo techniques, including sequential Monte Carlo (SMC) approaches (Lew et al., 2023; Anaya Gonzalez et al., 2025), sample from constrained distributions by generating multiple candidates using variants of constrained decoding (the static method) and selecting valid ones or resampling partial sequences. AWRS (Lipkin et al., 2025) uses adaptive weighted rejection sampling as a proposal distribution within SMC, tracking rejection statistics to compute importance weights and resampling particles to avoid dead ends. These methods converge to the constrained distribution in the limit, but have no principled stopping criterion, and can be highly inefficient.

Other approaches combine LMs with auxiliary probabilistic models to enforce constraints, e.g., GeLaTo (Zhang et al., 2023) or Ctrl-G (Zhang et al., 2024), often using DFAs or HMMs. These methods use surrogate models, are restricted to specific constraint classes, require additional training, and cannot guarantee exact sampling. Approximate inference methods such as Feynman–Kac Transformers (Qin et al., 2022; Lew et al., 2023) share similar limitations.

## 6 CONCLUSION

We introduced *Constrained Adaptive Rejection Sampling* (CARS), a principled extension of Adaptive Rejection Sampling for constrained decoding. Unlike prior methods that either rely on inefficient rejection sampling or approximate the target distribution via MCMC-style procedures, CARS always produces samples from the exact constrained distribution while adaptively pruning entire families of invalid continuations. This combination of fidelity and efficiency makes CARS when generating diverse, constraint-satisfying samples is critical—e.g., program fuzzing.

## ETHICS STATEMENT

This work adheres to the ICLR Code of Ethics. We identify no significant ethical concerns with our research:

**Data and Privacy:** Our evaluation uses publicly available datasets including Spider Text-to-SQL tasks, SyGuS synthesis tasks, SMILES molecular representations, and PDDL planning domains from Pyperplan. No personal or sensitive information is involved.

**Conflicts of Interest:** The authors declare no conflicts of interest or competing financial interests related to this work.

**LLM Usage:** In accordance with ICLR 2026 policies, we disclose that large language models were used for writing assistance, including grammar checking and text formatting.

## REPRODUCIBILITY STATEMENT

We have made substantial efforts to ensure the reproducibility of our results:

**Implementation:** The repository will include Python scripts for: (1) collecting the sampling traces for all experiments; and (2) evaluating the results for all tasks. We implemented our CARS framework as an extension of the Transformers-GAD and MCMC libraries (Park et al., 2024; Anaya Gonzalez et al., 2025). We use the llguidance (AI, 2025) implementation of GCD.

**Datasets:** All evaluation datasets are publicly available: SQL tasks from the Spider dataset, SyGuS tasks, SMILES exemplars from open chemical databases, and PDDL domains from the Pyperplan package.

**Experimental Setup:** Detailed experimental protocols are provided in Section F.3 and Section F.4, including specific software versions, hardware configurations (Section B), and parameter settings. All hyperparameters and random seeds used are explicitly specified.

**Models:** We use only publicly available language models with exact version specifications and commit hashes provided for reproducibility.

The combination of open-source code, public datasets, and detailed documentation should enable full replication of our experimental results.

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

# Appendix

Complete experimental details, additional results, and implementation specifics are provided in the appendix sections following the references, including:

- Complete hardware and software specifications (Section B)
- Complete prompts and grammar specifications for all benchmarks (Section F.2)
- Detailed fuzzing protocol and environment setup (Section F.3).
- Coverage measurement methodology via LLVM instrumentation (Section F.4).
- Analysis of severe distributional misalignment cases (**??**).
- Additional experimental results and ablation studies (**??**).

## A  DECLARATION OF LLM USAGE

Large Language Models (LLMs) are the object of study in this work. However, no LLM was used as a component of our core proposed methodology, or for any part of the experimental data analysis. We used ChatGPT as a writing assistant throughout the research process. Its use included refining prose, generating explanatory text for concepts, drafting document outlines, creating figure captions, and assisting with the generation of boilerplate code for data processing and plotting. All final claims, experimental designs, results, and conclusions were conceived and verified by the human authors, who take full responsibility for the scientific content of this paper.

## B  HARDWARE AND SOFTWARE

Our experiments were conducted on Ubuntu 22.04 LTS nodes with Intel Xeon Gold 6230 CPUs (2.10 GHz, 10 cores, 20 threads allocated) and 384 GB RAM. For GPU-accelerated workloads, we provisioned 6x NVIDIA RTX A6000 GPUs. Our implementation is based on Python 3.10.12, PyTorch 2.8.0 with CUDA 12.8, Transformers 4.55.4 and llguidance 0.7.30. For domain-specific experiments, we additionally used: AFL++ 4.00c, LLVM 14.0.0 for fuzzing, RDKit 2025.3.6 for molecular validity checking (SMILES), pyperplan 2.1 and Validate V4 for PDDL planning.

## C  HYPERPARAMETERS

For all language model decoding, we set the temperature to 1.0, top-p to 1.0, and top-k to 0 to allow sampling from the full token vocabulary without distributional distortion. We set the maximum number of newly generated tokens as follows:

- **Program fuzzing**: 512 tokens (JSON, XML, SQL)
- **Molecular generation (SMILES)**: 256 tokens
- **PDDL planning**: 128 tokens (Blocksworld), 256 tokens (Satellite), 1024 tokens (Depot)
- **SyGuS Benchmarks by Park et al. (2024)**: 512 tokens

## D  MODEL CHECKPOINT

We evaluate on two instruction-tuned models representing different architectural families:

- **Llama-3.1-8B-Instruct** (Grattafiori et al., 2024): `https://huggingface.co/meta-llama/Llama-3.1-8B-Instruct` (commit `0e9e39f`)
- **Qwen/Qwen2.5-7B-Instruct** (Qwen et al., 2025): `https://huggingface.co/Qwen/Qwen2.5-7B-Instruct` (commit `a09a354`)

Both models use BF16 precision with their default tokenizers and system prompts unchanged.

# E   COMPUTATIONAL OVERHEAD ANALYSIS

We provide an analysis of CARS's computational overhead by profiling across all benchmark tasks. We demonstrate that CARS's trie-based tracking mechanism incurs minimal overhead while maintaining efficiency gains.

## E.1   PROFILING METHODOLOGY

We conducted profiling across 37 benchmark runs spanning five domains (SyGuS, program fuzzing, SMILES generation, PDDL planning, and Text-To-SQL), collecting 4,000 successful samples over 24 hours of total runtime. For each run, we tracked:

- Wall-clock time breakdown by operation type
- Memory usage for trie storage (CPU and GPU)
- Trie statistics (nodes, depth, branching factor, reuse rate)
- Operation counts per sample

All profiling was conducted on the hardware described in Appendix B using a custom `Profiler` class that instruments CARS's key operations:

- **Inference timing:** Measured via `time.time()` wrapped around model forward passes
- **Trie operations:** Trie lookups, insertions, and recomputations with per-operation timing
- **Memory profiling:** CPU memory via `psutil`, GPU memory via `torch.cuda.memory_allocated()`
- **Trie structure:** Recursive traversal to compute nodes and their depth

## E.2   RUNTIME BREAKDOWN

Table 3: Runtime breakdown per successful sample averaged across 4,000 samples.

| Operation | Total Time (s) | Percentage |
|---|---|---|
| LLM Inference | 64,500 | 66.5% |
| Constraint Checking | 29,000 | 29.9% |
| Trie Operations (CARS-specific) | 325 | 0.3% |
| Other (I/O, logging, etc.) | 3,100 | 3.2% |
| **Total** | **96,925** | **100%** |

Table 3 shows the average time allocation per successful sample across all benchmarks. We observe,

- **Trie operations** account for only 0.3% of total runtime (median: 0.3%, range: 0.0–1.0% across runs)
- **LLM inference** dominates at 66.5%, inherent to all sampling-based methods
- **Constraint checking** (29.9%) includes probability reweighting and vocabulary masking, required by all constrained decoding methods (GCD, MCMC, CARS)

Per successful sample, CARS spent an average of 24 seconds total. The trie operations contributing to CARS's 0.3% overhead consist of:

- **Trie lookups:** $O(1)$ amortized per token
- **Trie insertions:** $O(|w|)$ per sample for new prefixes
- **Probability propagation:** $O(|w|)$ per sample for updating parent nodes

Table 4: Trie memory usage statistics across 36 benchmark runs.

| Statistic | Memory (MB) |
|---|---:|
| Median | 431 |
| Mean | 8,540 |
| Min | 169 |
| Max | 285,000 |
| 75th percentile | 2,250 |
| 90th percentile | 12,500 |

### E.3 MEMORY USAGE

Table 4 summarizes trie memory consumption across benchmarks.

The maximum memory usage (285GB) occured in the PDDL Satelite domain. For the remaining 39 runs, memory ranged from 169MB to 41GB with median 431MB.

**Freezing the trie.** For memory-constrained applications, users can freeze the trie after collecting a finite number of samples, $N$ and continue sampling from the frozen $R^{\mathcal{W}}$ distribution. This approach bounds the memory while still providing exact sampling with practical rejection rates. For example, freezing after 100 samples in the PDDL satelite domain case would cap memory at $\sim$2.5GB while maintaining 70%+ of the eventual acceptance rate improvement it would get. In practice, most of the samples that are "helpful" in reducing the rejection rate are discovered in early steps.

### E.4 TRIE REUSE STATISTICS

CARS achieves a **70.4% average trie reuse rate** (Table 5), meaning 70.4% of token decisions reuse cached constraint computations rather than querying the constraint checker. This demonstrates that the trie effectively amortizes constraint-checking costs across samples.

Table 5: Trie reuse statistics showing percentage of token decisions using cached results.

| Metric | Reuse Rate (%) |
|---|---|
| Mean | 70.4 |
| Median | 76.5 |
| Min | 23.1 |
| Max | 94.2 |

### E.5 COMPARISON WITH BASELINE METHODS

The 0.3% trie overhead is CARS-specific and represents the algorithmic cost of achieving exact sampling with monotonically improving efficiency. In comparison,

- **RS**: No additional overhead beyond LLM inference and constraint checking, but wastes computation on repeated invalid sequences
- **ARS**: Maintains a hash set of rejected prefixes with O(1) lookup overhead per token (negligible), but still higher rejection rates than CARS as it only learns from complete rejected sequences
- **RSFT**: Maintains a hash set of length-1 invalid prefixes with O(1) lookup overhead (negligible), but only prevents invalid first tokens
- **GCD**: No trie overhead, but produces biased samples by greedily masking invalid tokens
- **AWRS**: Combines ARS overhead with SMC particle management (tracking importance weights, resampling particles)
- **MCMC**: No trie overhead, but requires multiple Metropolis-Hastings correction steps per sample



```
You are an expert XML generator.
Make sure you generate valid and diverse XML.

Question 1:
Generate a short, valid and complex XML file.

Solution 1:
<?xml version="1.0" encoding="UTF-8"?>
<!DOCTYPE note [
  <!ELEMENT note (#PCDATA)>
]>
...

Question 2:
Generate a short, valid and complex XML file.

Solution 2:
<?xml version="1.0" encoding="UTF-8"?>
<!DOCTYPE status [
...

Question 3:
Generate a short, valid and complex XML file.

Solution 3:
```

```
document ::=
    PROLOG doctype_decl element

PROLOG ::=
    "<?xml" attribute* "?>"

doctype_decl ::=
    "<!DOCTYPE" NAME internal_dtd ">"

internal_dtd ::=
    "[" element_decl+ attlist_decl+ "]"

element_decl ::=
    "<!ELEMENT" NAME content_spec ">"

...

attribute ::=
    NAME "=" ESCAPED_STRING

content ::=
    (element | TEXT | cdata)*

cdata ::=
    "<![CDATA[" any_text "]]>"
```



(a) Prompt        (b) Grammar

Figure 4: (a) Prompt given to a LM to generate seed test cases for fuzzing the XML parser. (b) Simplified version of the XML grammar written in Lark notation. The goal of the problem is to generate multiple diverse seeds that trigger different code paths in the library being tested.

The net result is that CARS's 0.3% trie overhead enables 2–10× reduction in total samples needed (see Section 4), leading to substantial overall speedups despite the marginal computational cost.

## F FUZZING EXPERIMENTS

### F.1 BENCHMARKS

Table 6 summarizes the libraries, versions, and seed formats for each target. We note that, for our XML benchmark, our grammar targets libxml2's DOCTYPE/DTD parsing functionality, representing approximately 10-15% of the library's overall codebase.

Table 6: Fuzzing benchmarks, versions, and seed formats.

| Target | Library | Version | Seed format |
|---|---|---|---|
| XML[Group (2008; 2009)] | libxml2 | 2.15.0 | .xml |
| SQL[zxteloiv (2025)] | sqlite | 3.50.4 | .test |
| JSON[jso] | json-c | 0.18 | .json |

### F.2 PROMPTS AND CONSTRAINTS

For all benchmarks, we use a standard in-context learning format where the prompt consists of two (specification, solution) pairs, followed by a new specification for which the model must generate a solution. A representative prompt for the XML benchmark is shown in Figure 4a. In the "Prompts with Grammar" condition, this same prompt is augmented with the formal grammar specification shown in Figure 4b, while in the "Prompts without Grammar" condition, only the prompt examples are provided.

### F.3 FUZZING PROTOCOL AND ENVIRONMENT

All fuzzing experiments were conducted using AFL++ 4.00c on the hardware and software setup described in Section B. Each (benchmark, method) pair was evaluated in $N = 5$ independent, single-instance AFL++ runs of exactly 3600 s (one hour). We set 'AFL_RANDOM_SEED' to $42 + i, (i =$

1...5) for reproducibility and configure standard environment variables to ensure non-interactive execution. All other AFL++ parameters remained at defaults to isolate the impact of seed corpus quality. Complete builds and execution scripts are provided in the supplementary materials.

## F.4  COVERAGE MEASUREMENT VIA LLVM INSTRUMENTATION

We measured line coverage using LLVM's instrumentation toolchain with flags `-fprofile-instr-generate -fcoverage-mapping`, which adds $\leq 2\%$ runtime overhead. Raw profiles were collected during execution and aggregated post-trial using `llvm-profdata` and `llvm-cov`.

**Rationale.**  We report line coverage rather than crash counts because the experiment isolates *seed quality*—all methods receive identical prompts per benchmark, making coverage a direct measure of how effectively their generated seeds exercise the target code.

## F.5  COMPLEX CASE

We document a severe distributional misalignment scenario where even improved rejection sampling methods face fundamental limitations. This case occurs when the grammar constraints mandate syntactic elements that are absent from both the LM's training distribution and the prompt context.

**Experimental Setup.**  Initially, we tested SQL constraints in line with those used by Anaya Gonzalez et al. (2025), requiring mandatory `set ::timeout 60000` directives in every `.test` file. This syntax is severely misaligned with typical SQL in LM training data.

**Results.**  Across 2000 attempted samples using prompts *without* relevant examples:

- **Standard Rejection Sampling**: 0% acceptance rate.
- **CARS**: $< 0.1\%$ acceptance rate, always times out before reaching 100 valid samples.

This experiment shows that if the LLM is completely misaligned with the target constraint, our approach will not necessarily help. This phenomenon is an expected limitation of rejection sampling and in such settings one should opt for an inexact approach. To enable meaningful comparison between exact and approximate methods, we instead use the SQLite test-script grammar with mandatory `do_test` blocks—a challenging but feasible benchmark where exact methods remain viable.

## F.6  RESULTS

This section provides comprehensive fuzzing results across all benchmarks and conditions, complementing the representative results shown in Section 4.1. We present results for three grammar-intensive targets (JSON, SQL, XML) across two models (Llama-3.1-8B-Instruct, Qwen-2.5-7B-Instruct) under both prompt conditions (with/without grammar specification).

**Sample Efficiency Summary.**  Table 7 shows the number of LM generations required to produce 100 valid samples across all experimental conditions. Methods that timeout within the 2000-sample budget are marked as such.

**Line Coverage Results.**  Tables 8, 9, and 10 show downstream fuzzing performance measured by line coverage achieved after 1 hour of AFL++ execution.

**KL Divergence.**  Figure 5 shows distributional fidelity across benchmarks and conditions, measured as KL divergence from the empirically estimated target distribution.

**Key Findings.**  The comprehensive results reveal several important patterns across our three fuzzing benchmarks,

Table 7: Sample efficiency across fuzzing benchmarks—generations required for 100 valid samples.

| Method | JSON Llama | JSON Qwen | SQL Llama | SQL Qwen | XML Llama | XML Qwen |
|---|---|---|---|---|---|---|
| *Without Grammar in Prompt* | | | | | | |
| RS | T.O. | T.O. | T.O. | T.O. | T.O. | T.O. |
| ARS | T.O. | T.O. | T.O. | T.O. | T.O. | ~1253 |
| RSFT | ~601 | ~341 | ~1960 | T.O. | ~893 | ~1601 |
| CARS | ~130 | ~230 | ~1004 | ~1240 | ~440 | ~442 |
| GCD | 100 | 100 | 100 | 100 | 100 | 100 |
| AWRS | T.O. | T.O. | INFEASIBLE | INFEASIBLE | INFEASIBLE | INFEASIBLE |
| MCMC | 1000 | 1000 | 1000 | 1000 | 1000 | 1000 |
| *With Grammar in Prompt* | | | | | | |
| RS | T.O. | T.O. | T.O. | T.O. | T.O. | T.O. |
| ARS | T.O. | T.O. | T.O. | T.O. | T.O. | 612 |
| RSFT | ~874 | ~127 | ~1560 | T.O. | ~275 | ~731 |
| CARS | ~475 | ~131 | T.O. | T.O. | ~215 | ~548 |
| GCD | 100 | 100 | 100 | 100 | 100 | 100 |
| AWRS | T.O. | T.O. | INFEASIBLE | INFEASIBLE | INFEASIBLE | INFEASIBLE |
| MCMC | 1000 | 1000 | 1000 | 1000 | 1000 | 1000 |

Table 8: Line coverage results for JSON fuzzing benchmarks. Values show mean lines covered $\pm$ 95% CI over 5 independent trials.

| Method | Without Grammar Llama-3.1-8B-Ins. | Without Grammar Qwen-2.5-7B-Ins. | With Grammar Llama-3.1-8B-Ins. | With Grammar Qwen-2.5-7B-Ins. |
|---|---|---|---|---|
| RS | T.O. | T.O. | T.O. | T.O. |
| ARS | T.O. | T.O. | T.O. | T.O. |
| RSFT | $3,120 \pm 40$ | $3,050 \pm 30$ | $3,080 \pm 50$ | $3,060 \pm 40$ |
| CARS | $3,230 \pm 50$ | $3,090 \pm 40$ | $3,180 \pm 60$ | $3,070 \pm 30$ |
| GCD | $2,870 \pm 30$ | $2,850 \pm 20$ | $2,890 \pm 40$ | $2,860 \pm 30$ |
| AWRS | INFEASIBLE | INFEASIBLE | INFEASIBLE | INFEASIBLE |
| MCMC | $3,100 \pm 40$ | $3,190 \pm 50$ | $3,150 \pm 40$ | $3,170 \pm 60$ |

- **Method Feasibility and Timeout Patterns**—Rejection sampling methods (RS, ARS) consistently timeout across all benchmarks and conditions, confirming the computational intractability of naive approaches for constrained generation. AWRS proves to be computationally infeasible for all tested scenarios, for the infrastructure used in Section B highlighting the limitations of existing weighted approaches for complex constraint satisfaction.

- **CARS Performance Superiority**—Where feasible, CARS achieves the highest line coverage across most conditions. For JSON benchmarks, CARS reaches 32.3% coverage without grammar (vs. 28.7% for GCD), representing a 12.5% improvement. In XML fuzzing, CARS consistently achieves 9.7-9.8% coverage, outperforming all baselines including MCMC's 9.3-9.7% range.

- **Distributional Fidelity Translates to Coverage Quality**—The superior downstream fuzzing performance of CARS-generated seeds demonstrates that maintaining distributional fidelity under constraints yields tangible benefits in exploration diversity. Across benchmarks, CARS consistently outperforms approximate methods like GCD by 3-4%, confirming that exact sampling methods provide meaningful advantages for seed generation in fuzzing applications.

Table 9: Line coverage results for SQL fuzzing benchmarks. Values show mean lines covered $\pm$ 95% CI over 5 independent trials.

| Method | Without Grammar | | With Grammar | |
| | Llama-3.1-8B-Ins. | Qwen-2.5-7B-Ins. | Llama-3.1-8B-Ins. | Qwen-2.5-7B-Ins. |
|---|---|---|---|---|
| RS | T.O. | T.O. | T.O. | T.O. |
| ARS | T.O. | T.O. | T.O. | T.O. |
| RSFT | $22{,}223 \pm 394$ | T.O. | $20{,}174 \pm 473$ | T.O. |
| CARS | $22{,}456 \pm 315$ | $21{,}278 \pm 473$ | T.O. | T.O. |
| GCD | $20{,}726 \pm 236$ | $20{,}016 \pm 236$ | $19{,}700 \pm 315$ | $20{,}488 \pm 394$ |
| AWRS | INFEASIBLE | INFEASIBLE | INFEASIBLE | INFEASIBLE |
| MCMC | $21{,}041 \pm 315$ | $21{,}435 \pm 394$ | $19{,}858 \pm 236$ | $21{,}120 \pm 315$ |

Table 10: Line coverage results for XML fuzzing benchmarks. Values show mean lines covered $\pm$ 95% CI over 5 independent trials.

| Method | Without Grammar | | With Grammar | |
| | Llama-3.1-8B-Ins. | Qwen-2.5-7B-Ins. | Llama-3.1-8B-Ins. | Qwen-2.5-7B-Ins. |
|---|---|---|---|---|
| RS | T.O. | T.O. | T.O. | T.O. |
| ARS | T.O. | $6{,}764 \pm 267$ | T.O. | $6{,}764 \pm 178$ |
| RSFT | $8{,}366 \pm 356$ | $8{,}188 \pm 445$ | $8{,}544 \pm 267$ | $8{,}633 \pm 356$ |
| CARS | $8{,}814 \pm 267$ | $8{,}277 \pm 356$ | $8{,}633 \pm 445$ | $8{,}722 \pm 267$ |
| GCD | $7{,}117 \pm 178$ | $7{,}209 \pm 267$ | $8{,}544 \pm 356$ | $7{,}120 \pm 178$ |
| AWRS | INFEASIBLE | INFEASIBLE | INFEASIBLE | INFEASIBLE |
| MCMC | $8{,}764 \pm 356$ | $8{,}277 \pm 267$ | $8{,}633 \pm 178$ | $8{,}633 \pm 445$ |

# G  MOLECULAR GENERATION (SMILES)

## G.1  EXPERIMENTAL SETUP

We evaluate on three molecular classes from prior work Wang et al. (2023); Guo et al. (2022), representing distinct industrial chemical applications:

**Acrylates** (32 example molecules): Vinyl ester compounds for polymer synthesis, characterized by the `C=CC(=O)O` motif. **Chain Extenders** (11 example molecules): Difunctional molecules for polymer chain extension with hydroxyl or amine groups. **Isocyanates** (11 example molecules): Reactive compounds with `N=C=O` groups for polyurethane synthesis.

Each class employs hierarchical SMILES grammars enforcing both syntactic validity (balanced parentheses, valid bonds, ring closures) and semantic constraints (required functional groups). Figure 7 illustrates the prompt structure and grammar for acrylates; similar constructions apply to other classes.

## G.2  PARSE-TREE ILLUSTRATION

Figure 6 illustrates the parse tree for a representative Acrylate molecule (`C=CC(=O)OCC`). The tree demonstrates how the grammar enforces both SMILES syntax and the required acrylate functional group. Purple nodes represent non-terminals, green nodes show grammar terminals, and blue text displays the actual SMILES tokens.

## G.3  PROMPTS AND CONSTRAINTS

We use few-shot prompting where all available exemplars for each class serve as context. Figure 7a shows the prompt structure for Acrylates. The model receives all 32 known acrylates as examples, then must generate novel molecules satisfying the grammar in 7b.

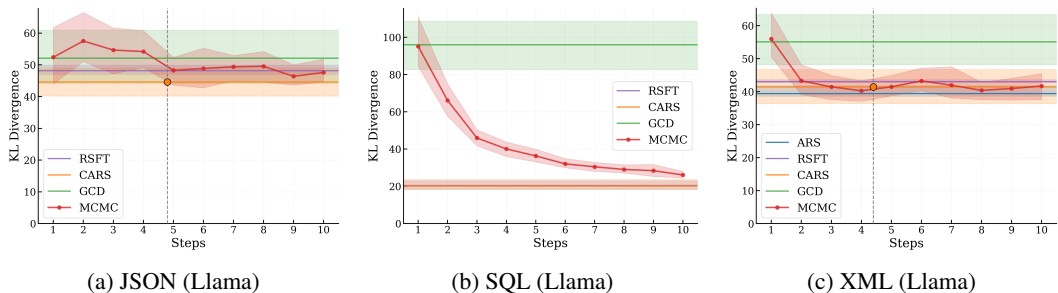

| (a) JSON (Llama) | (b) SQL (Llama) | (c) XML (Llama) |

Figure 5: KL divergence comparison across fuzzing benchmarks (without grammar condition). CARS and RSFT show consistently lower divergence than approximate methods, confirming distributional fidelity while MCMC shows convergence behavior over steps.

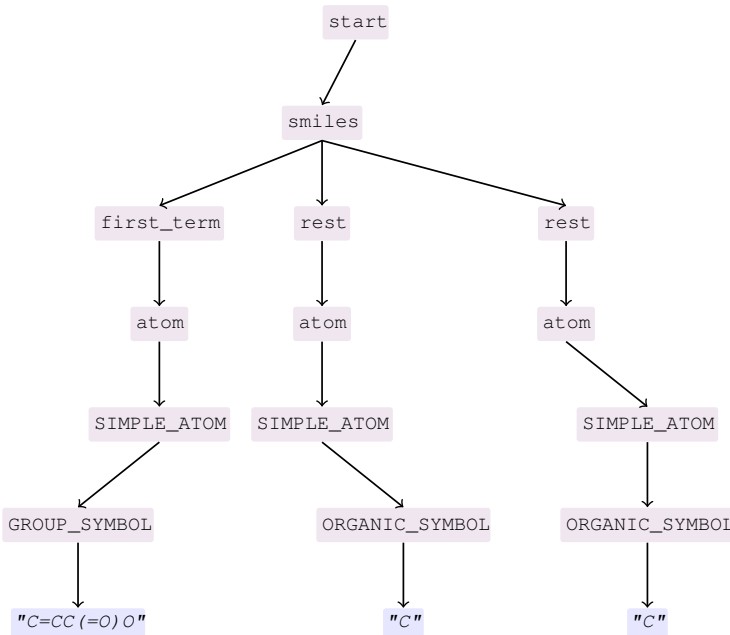

Figure 6: Parse tree for ethyl acrylate (C=CC(=O)OCC). The tree shows how grammar constraints enforce the acrylate functional group (highlighted) while permitting variation in alkyl substituents. Purple nodes represent non-terminals, and *blue italics* text displays the actual SMILES tokens.

## G.4 EVALUATION METRICS

We assess four complementary aspects of molecular generation quality,

**Validity.** Fraction of generated SMILES successfully parsed by RDKit RDKit without errors. This measures basic chemical plausibility.

**Diversity.** Average pairwise Tanimoto distance computed over Morgan fingerprints Rogers & Hahn (2010) with radius 2 and 2048 bits:

$$D = \frac{2}{n(n-1)} \sum_{i<j} (1 - T(M_i, M_j))$$

where $T$ is Tanimoto similarity and $M_i$ are molecular fingerprints. Higher values indicate more diverse chemical space coverage.

```
You are an expert in chemistry. You
are given several examples of acrylates
molecules in SMILES format. Your task is
to provide one new, valid acrylates
molecule in the SMILES format. Your
response must be a single SMILES
molecule and nothing else.

Molecule: C=CC(=O)OCC1=CC=CC=C1
Molecule: C=CC(=O)OC1=CC=CC=C1
Molecule: CC(=C)C(=O)OC1=CC=CC=C1
Molecule: C=CC(=O)OCCC1=CC=CC=C1
Molecule: CCCCCCCCCOC(=O)C(=C)C
Molecule: CCC(C)OC(=O)C=C
Molecule: CC(=C)C(=O)OCC1=CC=CC=C1
Molecule: CCCOC(=O)C=C
Molecule: CC(C)COC(=O)C(=C)C
Molecule: CCCCCCCCCCCCCOC(=O)C(=C)C
Molecule: CCC(C)OC(=O)C(=C)C
... (16 molecules)
Molecule: CCCOC(=O)C(=C)C
Molecule: CC1CC(CC(C1)(C)C)OC(=O)C(=C)C
Molecule: CC(C)CCCCCCCOC(=O)C=C
Molecule: CCCOC(=O)C=C
Molecule: COCCOC(=O)C=C
Molecule:
```

(a) Prompt

```
start : smiles

smiles : first_term rest*

first_term : atom branch* RING_CLOSURE*

rest : BOND? (atom branch* RING_CLOSURE* |
RING_CLOSURE)
...

SIMPLE_ATOM :
  ORGANIC_SYMBOL | AROMATIC_SYMBOL |
  WILDCARD | GROUP_SYMBOL

BOND : "-" | "=" | "#" | "$" | ":" |
    "/" | "\" | "."

ORGANIC_SYMBOL : "Br" | "Cl" | "N" | "O" |
    "P" | "S" | ...

AROMATIC_SYMBOL : "b" | "c" | "n" | "o" | ..

GROUP_SYMBOL : "C=CC(=O)O" | ...
...

ISOTOPE : "1".."9" ("0".."9")? ("0".."9")?
```

(b) Grammar

Figure 7: Acrylate generation setup showing (a) few-shot prompting with all 32 class exemplars and (b) simplified grammar enforcing both SMILES syntax and acrylate functional groups.

**Retrosynthesis Score.** Synthesizability estimated via RetroStar Chen et al. (2020), which predicts reaction pathways to available building blocks. Scores range [0,1] with higher values indicating easier synthesis.

**Class Membership.** Fraction correctly classified into the target chemical class via SMARTS pattern matching for required functional groups.

**Sample Efficiency.** Mean number of LM forward passes required to obtain 100 valid, unique molecules (excluding prompt exemplars). We impose a 1000-sample timeout and average over 3 independent trials.

**Note on drug-likeness metrics.** We intentionally omit QED Bickerton et al. (2012) and Lipinski's Rule of Five Lipinski et al. (2001) as the industrial chemical classes used in our evaluation (polymers, coatings) are not intended for pharmaceutical applications. Such metrics would be inappropriate for evaluating polymer precursors and specialty chemicals.

G.5 RESULTS

Tables 11 to 13 show detailed performance breakdown by chemical class for Llama-3.1-8B-Instruct. The results reveal varying constraint difficulty across chemical families, with Chain Extenders showing the highest baseline validity rates and Isocyanates presenting the most challenging generation task.

From Tables 11 to 13 we see that,

- **Acrylates**: CARS achieves best validity (0.93) and diversity (0.81), demonstrating strong performance across both efficiency and quality metrics.
- **Chain Extenders**: This is the highest baseline performance commonly across methods. CARS maintains competitive performance.
- **Isocyanates**: The most challenging class with low validity rates. Yet, CARS and ARS tie for best validity (0.76)
- **AWRS Performance**: Poor across classes mainly because it consistently reaches timeout. Generates sequences to 256-token limit vs. 31-token average for other methods.

Table 11: Molecular generation results for Acrylates dataset (Llama-3.1-8B-Instruct). Values show mean $\pm$ standard deviation over 3 trials.

| Method | Validity | Diversity | Retro Score | Membership |
|---|---|---|---|---|
| RS | $0.88 \pm 0.01$ | $0.76 \pm 0.01$ | $0.73 \pm 0.02$ | $0.88 \pm 0.01$ |
| ARS | $0.91 \pm 0.01$ | $0.74 \pm 0.00$ | $\mathbf{0.75 \pm 0.04}$ | $0.90 \pm 0.01$ |
| RSFT | $0.87 \pm 0.04$ | $0.74 \pm 0.02$ | $0.66 \pm 0.04$ | $0.86 \pm 0.04$ |
| CARS | $\mathbf{0.93 \pm 0.03}$ | $\mathbf{0.81 \pm 0.01}$ | $0.71 \pm 0.04$ | $\mathbf{0.92 \pm 0.02}$ |
| GCD | $0.61 \pm 0.01$ | $0.76 \pm 0.01$ | $0.50 \pm 0.01$ | $0.69 \pm 0.01$ |
| AWRS | $0.02 \pm 0.01$ | $0.76 \pm 0.01$ | $0.00 \pm 0.00$ | $0.02 \pm 0.02$ |
| MCMC | $0.57 \pm 0.02$ | $0.78 \pm 0.02$ | $0.51 \pm 0.01$ | $0.71 \pm 0.02$ |

Table 12: Molecular generation results for Chain Extenders dataset (Llama-3.1-8B-Instruct). Values show mean $\pm$ standard deviation over 3 trials.

| Method | Validity | Diversity | Retro Score | Membership |
|---|---|---|---|---|
| RS | $\mathbf{0.95 \pm 0.02}$ | $\mathbf{0.87 \pm 0.01}$ | $\mathbf{0.58 \pm 0.02}$ | $0.90 \pm 0.02$ |
| ARS | $0.94 \pm 0.01$ | $0.87 \pm 0.00$ | $0.52 \pm 0.06$ | $\mathbf{0.91 \pm 0.01}$ |
| RSFT | $0.94 \pm 0.02$ | $0.87 \pm 0.01$ | $0.53 \pm 0.01$ | $\mathbf{0.91 \pm 0.04}$ |
| CARS | $0.93 \pm 0.00$ | $\mathbf{0.88 \pm 0.01}$ | $0.54 \pm 0.03$ | $\mathbf{0.91 \pm 0.01}$ |
| GCD | $0.91 \pm 0.00$ | $0.86 \pm 0.01$ | $0.57 \pm 0.01$ | $0.89 \pm 0.00$ |
| AWRS | $0.03 \pm 0.00$ | $0.87 \pm 0.01$ | $0.01 \pm 0.00$ | $0.03 \pm 0.00$ |
| MCMC | $0.92 \pm 0.00$ | $0.87 \pm 0.00$ | $0.58 \pm 0.01$ | $0.90 \pm 0.00$ |

Tables 14 to 16 show results for Qwen2.5-7B-Instruct by chemical class.

**Sample Efficiency Analysis**   Table 17 provides detailed sample efficiency breakdown, showing the number of generations required to produce 100 valid molecules across models and chemical classes.

## H   TEXT-TO-SQL GENERATION

Table 18 presents text-to-SQL generation results using Qwen2.5-7B-Instruct on the Spider development set. The trends are consistent with Llama-3.1-8B-Instruct results: exact methods achieve the highest execution accuracy while CARS maintains sample efficiency comparable to other methods.

Qwen2.5-7B-Instruct demonstrates better overall performance on Spider compared to Llama-3.1-8B-Instruct, with CARS achieving 0.595 execution accuracy (vs. 0.578 for Llama). The relative improvements remain consistent: CARS outperforms GCD by 5.4 percentage points and achieves higher accuracy than MCMC and AWRS while requiring $9.3\times$ fewer samples. Among exact methods, CARS uses $1.8\times$ fewer samples than RS and $1.3\times$ fewer than ARS while maintaining the highest accuracy.

## I   PDDL PLANNING

In this experiment, we consider a benchmark where the goal is not to sample many diverse outputs but to solve a concrete task. Our goal is to assess whether exact samples from a constrained distribution are more likely to solve a downstream task.

We evaluate on three PDDL (Planning Domain Definition Language) settings from Zuo et al. (2025); Wang et al. (2023): *Blocks World*, *Depot*, and *Satellite*. For each domain, we construct few-shot prompts using four ground-truth plans and test on four randomly sampled tasks, targeting 100 valid action plans with a 1000-sample cap. Results are averaged over three independent trials.

### I.1   BENCHMARKS

We evaluate on three classical planning domains from Zuo et al. (2025); Wang et al. (2023):

Table 13: Molecular generation results for Isocyanates dataset (Llama-3.1-8B-Instruct). Values show mean $\pm$ standard deviation over 3 trials.

| Method | Validity | Diversity | Retro Score | Membership |
|---|---|---|---|---|
| RS | $0.72 \pm 0.09$ | $\mathbf{0.87 \pm 0.01}$ | $0.45 \pm 0.08$ | $0.68 \pm 0.08$ |
| ARS | $\mathbf{0.76 \pm 0.06}$ | $0.86 \pm 0.00$ | $\mathbf{0.48 \pm 0.01}$ | $0.71 \pm 0.06$ |
| RSFT | $0.65 \pm 0.04$ | $0.86 \pm 0.00$ | $0.41 \pm 0.01$ | $0.63 \pm 0.04$ |
| CARS | $\mathbf{0.76 \pm 0.01}$ | $\mathbf{0.87 \pm 0.01}$ | $0.47 \pm 0.02$ | $\mathbf{0.72 \pm 0.03}$ |
| GCD | $0.64 \pm 0.00$ | $0.87 \pm 0.00$ | $0.32 \pm 0.00$ | $0.64 \pm 0.01$ |
| AWRS | $0.13 \pm 0.01$ | $0.81 \pm 0.10$ | $0.04 \pm 0.01$ | $0.03 \pm 0.00$ |
| MCMC | $0.67 \pm 0.00$ | $0.86 \pm 0.00$ | $0.37 \pm 0.00$ | $0.67 \pm 0.02$ |

Table 14: Molecular generation results for Acrylates (Qwen2.5-7B-Instruct). Values show mean $\pm$ standard deviation over 3 trials.

| Method | Validity | Diversity | Retro Score | Membership |
|---|---|---|---|---|
| RS | $0.92 \pm 0.01$ | $0.31 \pm 0.00$ | $0.92 \pm 0.02$ | $0.92 \pm 0.01$ |
| ARS | $\mathbf{1.00 \pm 0.00}$ | $0.23 \pm 0.01$ | $\mathbf{0.99 \pm 0.00}$ | $\mathbf{1.00 \pm 0.00}$ |
| RSFT | $\mathbf{1.00 \pm 0.00}$ | $0.16 \pm 0.03$ | $\mathbf{1.00 \pm 0.00}$ | $\mathbf{1.00 \pm 0.00}$ |
| CARS | $0.99 \pm 0.00$ | $\mathbf{0.25 \pm 0.02}$ | $0.97 \pm 0.01$ | $0.99 \pm 0.00$ |
| GCD | $0.70 \pm 0.01$ | $0.17 \pm 0.02$ | $0.60 \pm 0.03$ | $0.70 \pm 0.02$ |
| AWRS | $0.09 \pm 0.01$ | $\mathbf{0.72 \pm 0.02}$ | $0.01 \pm 0.03$ | $0.09 \pm 0.04$ |
| MCMC | $0.78 \pm 0.01$ | $0.25 \pm 0.01$ | $0.78 \pm 0.01$ | $0.78 \pm 0.02$ |

- **Blocks World** (4 tasks): Stacking and unstacking blocks to achieve goal configurations
- **Depot** (4 tasks): Logistics domain with trucks, hoists, and crates requiring coordinated movement and loading operations across multiple locations.
- **Satellite** (4 tasks): Satellite observation scheduling with actions for pointing instruments, calibrating, and taking images of celestial targets.

Each domain employs PDDL action grammars that enforce:

- **Syntactic validity**: Correct PDDL action syntax with proper operator names, parameter lists, and parenthesis matching.
- **Type constraints**: Parameters must match declared object types (e.g., `block`, `truck`, `satellite`).
- **Arity constraints**: Correct number of arguments for each action operator.

Figure 9b shows the grammar for Satellite actions. The grammar ensures syntactic correctness but does not enforce semantic constraints (preconditions/effects), which are verified separately during evaluation.

## I.2   PARSE-TREE ILLUSTRATION

Figure 8 illustrates the parse tree for a Satellite action sequence. The tree demonstrates how the grammar validates action syntax and parameter types.

## I.3   PROMPTS AND CONSTRAINTS

We use four-shot in-context learning where each example contains a PDDL problem specification (initial state and goal) paired with its ground truth action plan. The prompt includes the domain specification, which defines the available actions and object types. Figure 9a shows the prompt structure for Satellite.

Table 15: Molecular generation results for Chain Extenders (Qwen2.5-7B-Instruct). Values show mean $\pm$ standard deviation over 3 trials.

| Method | Validity | Diversity | Retro Score | Membership |
|---|---|---|---|---|
| RS | $0.98 \pm 0.00$ | $0.76 \pm 0.01$ | $0.31 \pm 0.02$ | $0.98 \pm 0.00$ |
| ARS | $0.99 \pm 0.00$ | $0.76 \pm 0.02$ | $\mathbf{0.38 \pm 0.04}$ | $0.99 \pm 0.00$ |
| RSFT | $0.99 \pm 0.00$ | $0.75 \pm 0.03$ | $0.31 \pm 0.01$ | $0.99 \pm 0.00$ |
| CARS | $0.99 \pm 0.00$ | $0.75 \pm 0.02$ | $0.37 \pm 0.01$ | $0.99 \pm 0.00$ |
| GCD | $\mathbf{1.00 \pm 0.00}$ | $0.71 \pm 0.04$ | $0.22 \pm 0.03$ | $\mathbf{1.00 \pm 0.00}$ |
| AWRS | $0.25 \pm 0.02$ | $\mathbf{0.87 \pm 0.01}$ | $0.11 \pm 0.02$ | $0.25 \pm 0.00$ |
| MCMC | $\mathbf{1.00 \pm 0.00}$ | $0.75 \pm 0.03$ | $0.30 \pm 0.04$ | $0.98 \pm 0.01$ |

Table 16: Molecular generation results for Isocyanates (Qwen2.5-7B-Instruct). Values show mean $\pm$ standard deviation over 3 trials.

| Method | Validity | Diversity | Retro Score | Membership |
|---|---|---|---|---|
| RS | $0.87 \pm 0.04$ | $0.76 \pm 0.03$ | $0.41 \pm 0.04$ | $0.87 \pm 0.03$ |
| ARS | $0.86 \pm 0.03$ | $0.76 \pm 0.02$ | $0.41 \pm 0.03$ | $0.86 \pm 0.02$ |
| RSFT | $\mathbf{0.91 \pm 0.02}$ | $0.77 \pm 0.01$ | $0.42 \pm 0.03$ | $\mathbf{0.91 \pm 0.02}$ |
| CARS | $0.87 \pm 0.02$ | $0.76 \pm 0.03$ | $\mathbf{0.43 \pm 0.02}$ | $0.87 \pm 0.02$ |
| GCD | $\mathbf{0.90 \pm 0.02}$ | $\mathbf{0.79 \pm 0.01}$ | $\mathbf{0.58 \pm 0.01}$ | $\mathbf{0.90 \pm 0.02}$ |
| AWRS | $0.03 \pm 0.01$ | $\mathbf{0.89 \pm 0.01}$ | $0.02 \pm 0.00$ | $0.03 \pm 0.00$ |
| MCMC | $\mathbf{0.90 \pm 0.02}$ | $\mathbf{0.79 \pm 0.01}$ | $\mathbf{0.51 \pm 0.01}$ | $\mathbf{0.90 \pm 0.02}$ |

## I.4 EVALUATION METRICS

Following Zuo et al. (2025); Loula et al. (2025), we employ evaluation metrics with increasing orders of strictness,

**Sample Efficiency.** Mean number of LM forward passes required to obtain 100 parseable plans. We impose a 1000-LM-generation timeout per task and average over 4 tasks for every domain, across 3 trials.

**Prefix Validity (PV).** Among parseable plans, this is the fraction of plans where the first 4 actions are: (1) Executable from the initial state (preconditions are satisfied);(2) Result in a state from which the goal is reachable (verified via search). This metric assesses semantic coherence and planning feasibility.

**Ground Truth Similarity (GTS).** Exact match rate between the first 4 generated actions and the reference solution. This measures alignment with expert planning strategies.

**Rationale for metrics.** PDDL generation from natural language is challenging - models frequently produce syntactically correct but semantically invalid plans - especially for problems with over 10 objects. The cascading framework distinguishes surface-level correctness (parsing) from deeper planning competence (executability, goal-directedness).

Natural language–to–PDDL generation is notoriously difficult: models often produce sequences that are syntactically malformed or semantically invalid. For semantic quality, we follow the cascading evaluation by Zuo et al. (2025); Loula et al. (2025) and measure the metrics above.

## I.5 RESULTS

Table 19 summarizes efficiency results across both models. The RS rates highlight strong variation in constraint alignment: Qwen2.5-7B-Instruct achieves moderate alignment (generating 38% valid samples), whereas Llama-3.1-8B-Instruct fails to produce 100 samples within the cap.

Table 17: Sample efficiency for molecular generation: generations required for 100 valid molecules.

| | Llama-3.1-8B-Instruct | | | Qwen2.5-7B-Instruct | | |
|---|---|---|---|---|---|---|
| Method | Acrylates | Chain Ext. | Isocyanates | Acrylates | Chain Ext. | Isocyanates |
| RS | $\sim$2100* | $\sim$105 | $\sim$139 | $\sim$3000* | $\sim$102 | $\sim$115 |
| ARS | $\sim$871 | $\sim$106 | $\sim$132 | $\sim$129 | $\sim$101 | $\sim$116 |
| RSFT | $\sim$2100* | $\sim$106 | $\sim$154 | $\sim$3333* | $\sim$101 | $\sim$110 |
| CARS | $\sim$**277** | $\sim$**108** | $\sim$**132** | $\sim$**112** | $\sim$**101** | $\sim$**115** |
| GCD | 100 | 100 | 100 | 100 | 100 | 100 |
| AWRS | 1000 | 1000 | 1000 | 1000 | 1000 | 1000 |
| MCMC | 1000 | 1000 | 1000 | 1000 | 1000 | 1000 |

*Extrapolated from low valid sample counts

Table 18: Text-to-SQL generation performance on Spider development set using Qwen2.5-7B-Instruct. Quality metrics show mean $\pm$ standard deviation over 4 trials. Bold indicates best performance.

| Method | Execution Accuracy | Total Samples | Samples/Query |
|---|---|---|---|
| RS | $\mathbf{0.593 \pm 0.012}$ | $2047 \pm 142$ | $\sim 1.98$ |
| ARS | $0.591 \pm 0.013$ | $1389 \pm 117$ | $\sim 1.34$ |
| RSFT | $0.589 \pm 0.010$ | $1852 \pm 163$ | $\sim 1.79$ |
| CARS | $\mathbf{0.593 \pm 0.011}$ | $1108 \pm 87$ | $\sim 1.07$ |
| GCD | $0.541 \pm 0.009$ | $\mathbf{1034 \pm 0}$ | $\mathbf{1.00}$ |
| AWRS | $0.582 \pm 0.016$ | $10340 \pm 0$ | $10.0$ |
| MCMC | $0.584 \pm 0.013$ | $10340 \pm 0$ | $10.0$ |

For Qwen2.5-7B-Instruct, CARS uses $1.2\times$ fewer LM calls than the other best exact method, ARS. For Llama-3.1-8B-Instruct, existing exact methods, RS and ARS, fail to produce 100 samples, while 61% of LM calls attempted by CARS produce valid samples.

The KL divergence of CARS is on average $2.1\times$ lower than MCMC and $2.8\times$ lower than AWRS.

Semantic quality is overall low, reflecting the inherent difficulty of generating PDDL. Nonetheless, exact methods slightly outperform approximate methods.

**Summary.** Across diverse PDDL domains, CARS is consistently more efficient than other exact methods, sometimes converting otherwise intractable sampling problems into feasible ones.

# J SyGuS Benchmarks by Park et al. (2024)

For completeness, we also evaluate CARS on the synthesis benchmarks introduced by Park et al. (2024). These tasks involve synthesizing expressions in an extension of linear integer arithmetic (SLIA) and loop invariants with bit-vector arithmetic (BV4). The problems are specified in the Syntax-Guided Synthesis (SyGuS) format (Alur et al., 2019), which provides both a logical specification and a context-free grammar of admissible terms. Following prior work, prompts consist of three in-context examples (specification–solution pairs), and the grammar is then given as a constraint for grammar-aligned sampling. The full benchmark contains 29 problems (14 BV4 and 15 SLIA).

While SyGuS is a natural setting for constrained generation, this benchmark is a somewhat imperfect fit for our problem formulation. The metric of interest here is the ability to produce *many diverse valid samples*, yet in real synthesis applications the key goal is to obtain *a single correct solution*. Thus, although we report results for completeness and comparability with prior work, we view this evaluation as secondary to the benchmarks in the main text.

**Setup.** We compare CARS against four rejection sampling variants (RS, ARS, RSFT, and our method CARS), MCMC-restart (Anaya Gonzalez et al., 2025) with $k \in \{1, \dots, 10\}$ steps (with greedy constrained decoding, GCD, being the case $k = 1$), and report three trials per method. Each

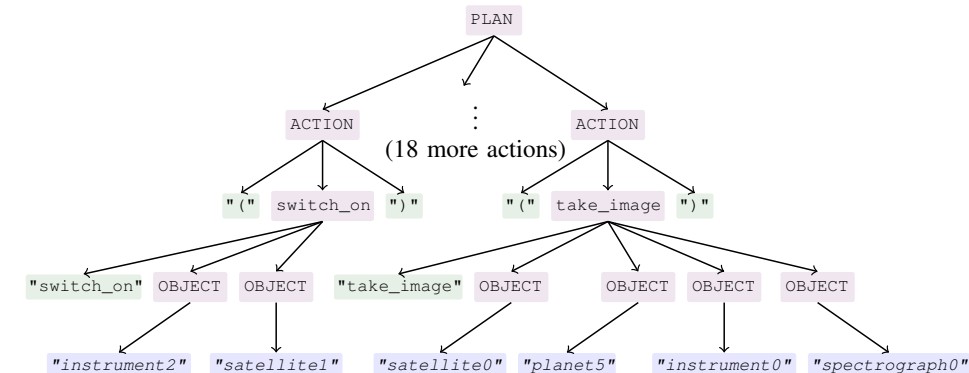

Figure 8: Parse tree for Satellite action sequence showing first *(switch_on instrument2 satellite1)* and last *(take_image satellite0 planet5 instrument0 spectrograph0)* actions, with 17 intermediate actions skipped for brevity. Purple nodes represent non-terminals, green boxes denote grammar terminals, and blue italic text displays the action parameters.

```
You are a PDDL planning expert. You are
given a domain, and some examples of
planning problems and a valid sequences
to achieve the goal.
...
Your final output must be a valid
sequence of actions.

Domain: SATELLITE

Domain Definition:
(define (domain satellite)
(:requirements :strips)
(:predicates
  (on_board ?i ?s) ... )
...
Problem:
(:objects
...
)
(:init
 (satellite satellite0)
...
 (direction Phenomenon7)
)
(:goal (and
 (pointing satellite0 Phenomenon5)
... (have_image Star4 spectrograph2)
...
))

Solution:
```

(a) Prompt

```
start : PLAN

PLAN :
    ACTION (" " ACTION)*

ACTION :
    "(" action_body ")"

action_body :
    binary_action " " OBJECT ...

binary_action :
    "switch_on" | "switch_off"

ternary_action :
    "turn_to" | "calibrate"

quaternary_action :
    "take_image"

OBJECT :
    "instrument" digit_0_7
    | "satellite" digit
    | "groundstation" digit
    | "phenomenon" digits
    | "planet" digits
    | "star" digits
...

digit_0_7 : "0".."7"
digit : "0".."9"
digits : digit+
```

(b) Grammar

Figure 9: (a) 4-shot prompt for Satellite planning. (b) Simplified version of the Satellite PDDL actions written in Lark notation. The grammar enforces correct action syntax for satellite manipulation operations.

trial generates 100 samples for each of the 29 problems, with a limit of 2000 LLM calls. If the limit of 2000 calls is reached, we report the number of samples produced within that limit. Because of the size of this benchmark, its secondary importance, and our limited computed budget we restrict evaluation to a single model: Llama-3.1-8B-Instruct.

**Efficiency.** Figure 10 reports the number of model calls required to generate 100 samples (each bar shows the median of 3 runs). Standard rejection sampling (RS) fails completely, often producing zero samples within the timeout. Restricting only the first token (RSFT) already helps substantially, since models otherwise tend to start with phrases like `The solution is` rather than a valid program.

Table 19: PDDL Planning results: sample efficiency and semantic quality metrics. In case of a timeout (–), we measure semantic quality on the <100 results produced before the timeout.

| Method | Qwen2.5-7B-Instruct | | | Llama-3.1-8B-Instruct | | |
|---|---|---|---|---|---|---|
| | % Valid | Prefix Validity | Gr. Truth Similarity | % Valid | Prefix Validity | Gr. Truth Similarity |
| RS | 38% | 4.0% | 1.2% | – | 0.2% | 0.0% |
| ARS | 54% | 4.3% | 0.9% | – | 0.7% | 0.0% |
| RSFT | 51% | **6.4**% | 1.9% | 36% | **3.0**% | **0.5**% |
| CARS | 66% | 6.3% | **2.5**% | 61% | 2.7% | **0.5**% |
| GCD | **100%** | 2.0% | 1.0% | **100%** | 1.0% | 0.0% |
| MCMC | **100%** | 2.6% | 1.7% | **100%** | 0.7% | 0.0% |
| AWRS | **100%** | 1.4% | 0.4% | **100%** | 1.0% | 0.1% |

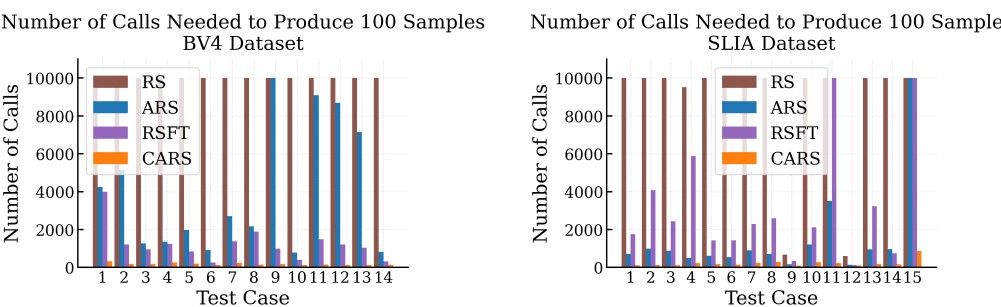

Figure 10: Number of of LLM calls required to produce 100 samples for BV4 and SLIA. Lower is better.

Still, CARS achieves order-of-magnitude improvements: on BV4, CARS uses $16\times$ fewer calls to the LLM (geomean) than ARS and $5.7\times$ fewer than RSFT; on SLIA, the corresponding factors are $4.5\times$ and $11.4\times$.

**Distributional Quality.** As a second evaluation metric, we measure the KL divergence defined in Section 4. Figures 11 and 12 shows KL divergences with $95\%$ confidence intervals (from bootstrapping). For MCMC-restart, the divergence depends on the number of steps $k$, plotted on the horizontal axis. As observed by Anaya Gonzalez et al. (2025), increasing $k$ generally reduces KL divergence, though with fluctuations due to randomness. For rejection-based methods (RS, ARS, RSFT, CARS), the theoretical KL divergence is $0$; however, empirical estimates from finite samples need not be exactly $0$.

To compare fairly, we also mark the computational budget of CARS (number of calls needed to generate 100 samples, from Figure 10) as a vertical on the MCMC curve. We see that in nearly all cases, the KL divergence of CARS is significantly lower than that of MCMC at comparable budget, highlighting that CARS delivers both efficiency and distributional faithfulness.

We note that to produce the first sample CARS usually needs more than two calls to the LLM, but what matters is that the amortized complexity of generating many samples becomes lower.

**Summary.** Although the SyGuS benchmarks are not directly aligned with the one-solution synthesis objective that motivates CARS, they nonetheless confirm the central message: *CARS transforms rejection sampling from essentially unusable into a highly efficient constrained generator, outperforming both prior rejection-based methods and MCMC baselines.*

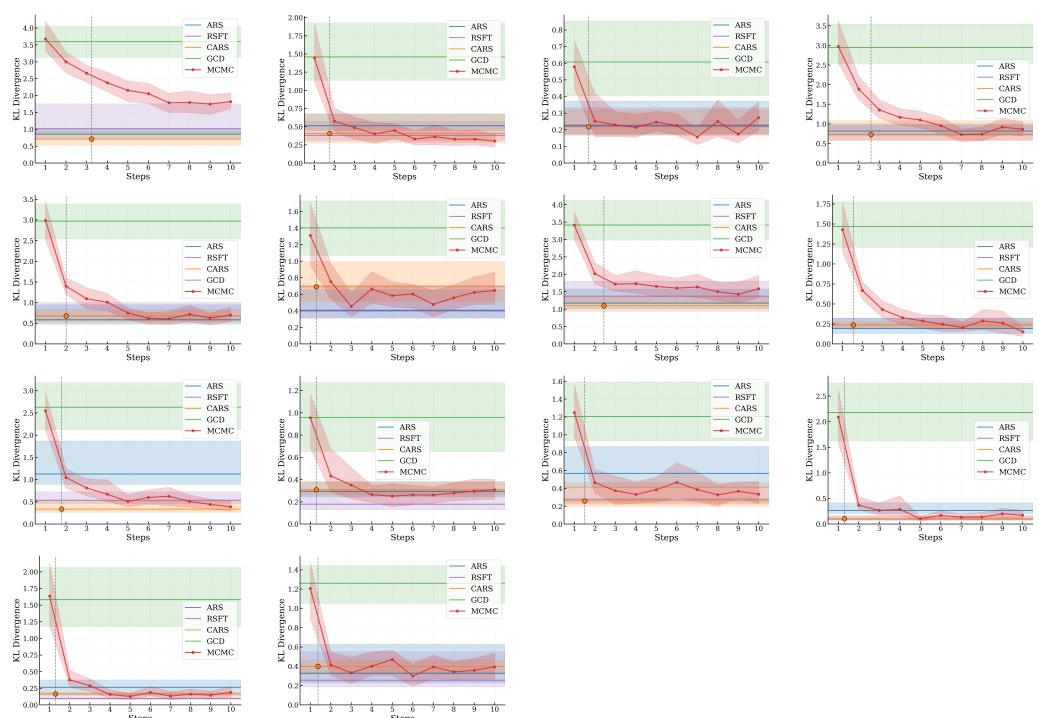

Figure 11: KL divergence for particular algorithms in BV4 subset

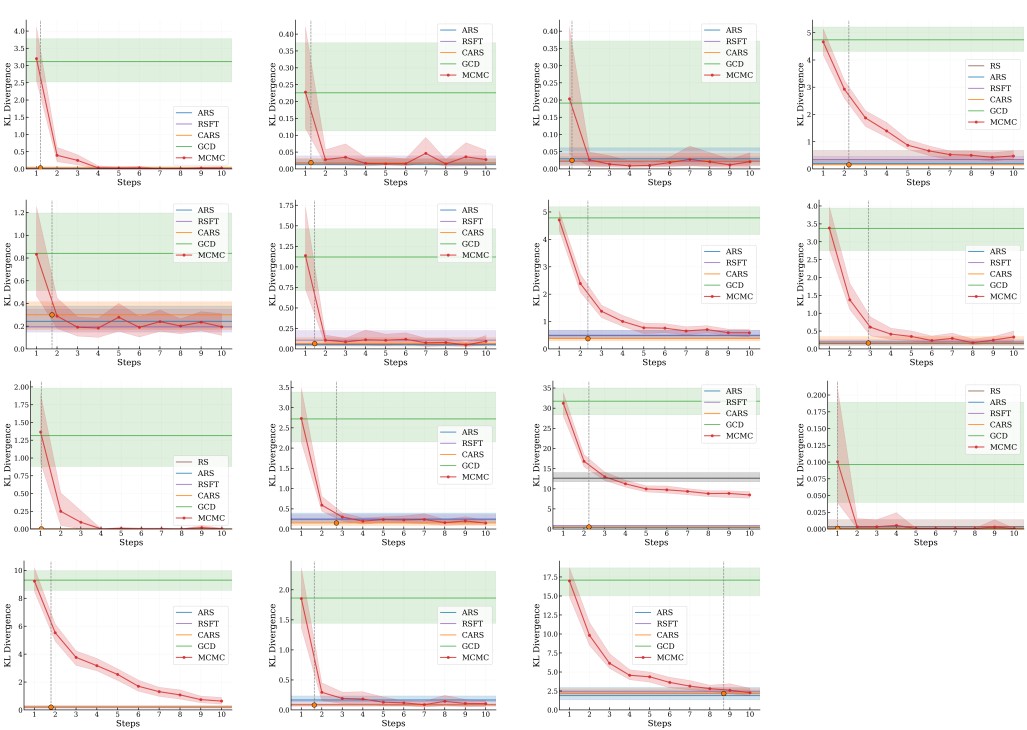

Figure 12: KL divergence for particular algorithms in SLIA subset

Additionally, we selected three representative problems from the BV4 dataset and ran a single CARS trial for each. Figure 13 shows the *success rate*—the number of valid samples produced divided by the

total number of LLM calls—as more calls are made. Note that the success rate is not monotonically increasing: any call that does not produce a valid sample temporarily decreases the rate. Figure 14 presents the same data in an alternative view, showing the cumulative number of valid samples generated within a given number of LLM calls.

The complete experimental results for the fuzzing approach with 1000 calls are presented in Figure 15, while Figure 16 shows the detailed progression of sample generation over time, including a zoomed view of the first 100 calls.

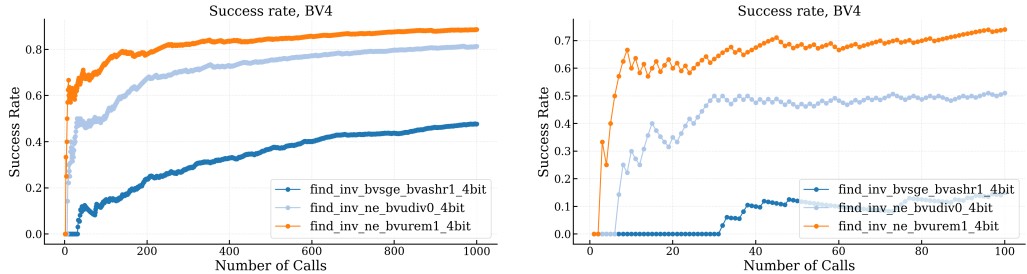

Figure 13: SyGuS: Dependence of the success rate on the number of calls (with a zoom to the first 100 calls)

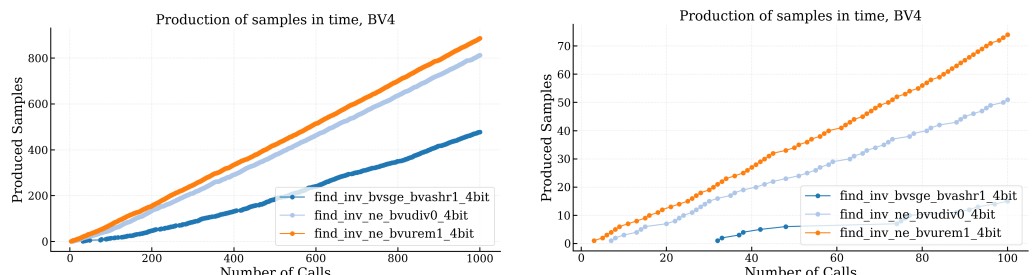

Figure 14: SyGuS: The number of produced samples as a function the number of calls (with a zoom to the first 100 calls)

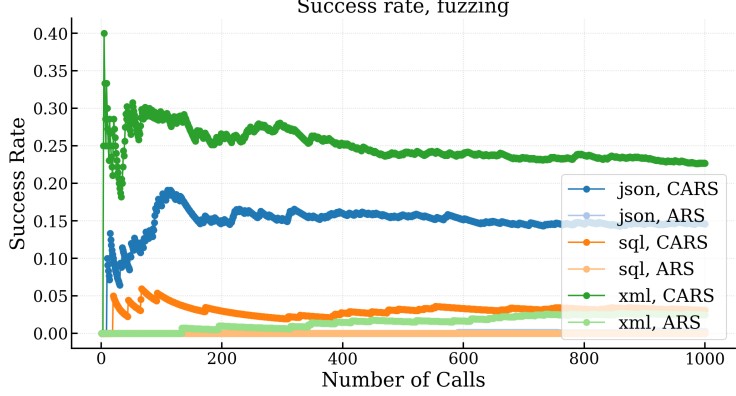

Figure 15: Fuzzing: Dependence of the success rate on the number of calls (1000 calls)

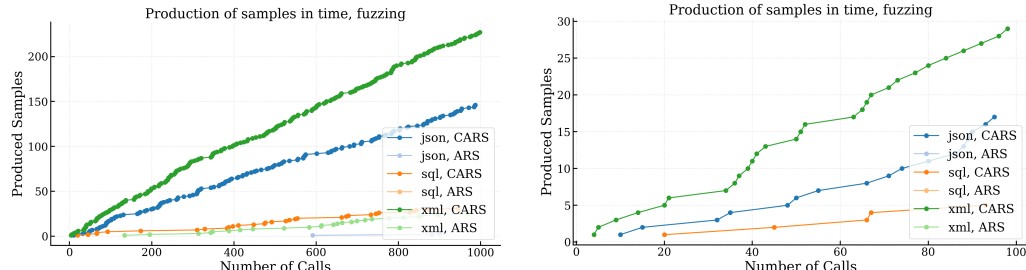

Figure 16: Fuzzing: The number of produced samples as a function the number of calls (with a zoom to the first 100 calls)

