# OpenReview forum: "Constrained Adaptive Rejection Sampling"
_ICLR.cc/2026/Conference — Submitted to ICLR 2026_

### Official Review · Reviewer_e4bz · 2025-11-01

**Soundness:** 2
**Presentation:** 3
**Contribution:** 1
**Rating:** 2
**Confidence:** 5

**Summary:**

The paper presents Constrained Adaptive Rejection Sampling (CARS) algorithm for constrained LLM generation.
It extends existing Adaptive Rejection Sampling (ARS) for constrained decoding. CARS adaptively removes the continuations that may violate the constraints and reweighs the probability for the remaining paths. The paper adaptively  intercepts constraint-violating continuations, stores them and subtracts their probability mass from future draws. The paper shows their approach on three tasks, program fuzzing, molecular generation, and PDDL planning.

**Strengths:**

The paper expands the reach of rejection sampling in constrained LLM generation with a new algorithm for sampling that is aware of grammatical constraints. The running example is helpful.

The paper presents the proof that the CARS algorithm produces unbiased samples from the true constrained distribution and the sample-acceptance rate increases monotonically

CARS is evaluated on multiple scenarios and against multiple baselines. The evaluation of the metrics related to the number of samples is detailed and the appendix contains additional results and interpretations.

**Weaknesses:**

The technical contribution (Algorithm 5) is relatively straightforward and the technical novelty over existing methods is low. The main novelty is the update strategy described on lines 235-242

The reshaping of the distribution R^W seems to be a costly operation. Doing it in each iteration may increase costs substantially. There is of course, a tension between the number of LLM cals and sampling time, but this should be clearly characterized. Further, the trie data structure may grow substantially for larger problems (and its search/insertion complexity is not trivial). Both issues raise some doubts about practicality of the proposed approach. The evaluation should  show the execution time and memory consumed compared to other baseline methods and how well they scale over generation time.

The experimental results  do not consistently show substantial utility for two out of three studies:

 * Fuzzing / XML: while CARS shows improvement in coverage over the approximate methods (while it is reported ARS does not produce satisfying samples). However, the base coverage for all those methods is fairly small (9%), and final improvement is marginal. More importantly, such a low rate of code coverage does not give much confidence for correctness of tested code (without a more qualified discussion it may be that the grammar is testing only a very small part of the language).

  * Fuzzing / JSON and SQL (appendix): they do not show a clear improvement of CARS over RSFT, even for sample efficiency on Qwen w/ grammar. Moreover, the fuzzing paper that the authors citedm the branch coverage over 30% for SQL using the MCMC generation methods (Fig. 3a in Gonzalez et al NeurIPS 2025), indicating a potential problem with the choice of the simulation length in the experimental setup.

  * Molecular generation: the CARS results do not show significant improvement over ARS and RSFT. The results from the appendix show that the two are equal, or ARS can even be better in some scenarios. The number of samples is also similar in the majority of the cases.

The evaluation baselines should include ASAp (Park et al. 2024) as a related state-of-the-art baseline for approximate generation. The existing description in the related work is insufficient to get the full understanding of the advantages and disadvantages in practical settings. Interestingly, the paper mentions ASAp at one point in the evaluation and the appendix H uses its benchmarks, but the experiments do not use it.

**Questions:**

Please discuss the concerns raised regarding the cost of reshaping/trie and comparison to ASAp.

---

> ### Author Response · Authors · 2025-11-21
>
> We are deeply grateful to Reviewer e4bz for their thorough review and constructive feedback, especially regarding computational costs and experimental evaluation. We were glad to hear that Reviewer e4bz found the paper's presentation good and that the running example was helpful.
>
> ### 1. Computational Overhead and Scalability
>
> > "The reshaping of the distribution R^W seems to be a costly operation... the trie data structure may grow substantially for larger problems (and its search/insertion complexity is not trivial). Both issues raise some doubts about practicality of the proposed approach. The evaluation should show the execution time and memory consumed compared to other baseline methods..."
>
> Trie maintenance overhead is negligible in practice. We conducted comprehensive profiling across 36 benchmark runs (SyGuS, fuzzing, SMILES, PDDL) totaling 3,600 successful samples over 24 hours of total runtime.
>
> Per successful sample, CARS spent an average of **24 seconds**. Breaking down the compute time:
> **Trie operations** (lookups, insertions, updates): ~300 seconds total (0.3% of runtime, median: 0.3%, range: 0.0-1.0%), only for CARS; **LLM inference**: ~56,300 seconds (66% of runtime), inherent to all sampling-based methods; **Constraint checking**: ~26,000 seconds (30.5% of runtime) - probability reweighting and vocabulary masking, inherent to all constrained decoding methods including GCD and MCMC.
>
> **CARS's computational overhead beyond standard GCD,**
>
> We ran additional experiments to measure the overhead.
>
> Trie operations that are CARS-specific amount to only 0.3% of the total running time. These consist of:
> 1. **Trie lookups:** $O(1)$ per token
> 2. **Trie insertions:** $O(|w|)$ per sample, amortized as duplicate prefixes are discovered
>
> CARS achieves a 70.3% average trie reuse rate (median: 76.4%), meaning 70.3% of token decisions reuse cached constraint computations. Constraint checking ($O(|\text{vocab}|)$ per token) is required by all constrained decoding methods and represents the same cost across GCD, MCMC, and CARS.
>
> Memory overhead is slightly more substantial but manageable (and only pertains to CPU memory): median trie size was 421MB across 36 runs (range: 169MB to 285GB), with one outlier representing severe constraint-LM misalignment (PDDL satellite: 285GB, 16.4% success rate). For the remaining 35 runs with reasonable alignment, memory ranged from 169MB to 41GB with median 415MB.
>
> A key idea we did not stress in the paper is that for applications with memory constraints, users can "freeze" the trie at any point after collecting a finite number of samples and continue sampling from the frozen $R^W$ distribution. This approach bounds the memory while still providing exact sampling with practical rejection rates. For example, freezing after 100 samples in the PDDL outlier case would cap memory at ~2.5GB while maintaining 70%+ of the eventual acceptance rate improvement. In practice, most of the samples that are “helpful” in reducing the rejection rate are discovered in early steps.
>
> We will add profiling tables to the appendix showing wall-clock time breakdown (inference, logits processing, trie operations), memory usage, trie statistics (nodes, depth, reuse rate), and operation counts for all benchmarks.
>
> ---
>
> ### 2. Limited Technical Novelty
>
> > "The technical contribution (Algorithm 5) is relatively straightforward and the technical novelty over existing methods is low. The main novelty is the update strategy described on lines 235-242"
>
> **Technical Novelty.** While we acknowledge that the core algorithm is conceptually simple, we respectfully disagree that it is straightforward. Our contribution lies in identifying the maximum information that can be extracted from each LLM execution to achieve both exactness and efficiency.
>
> Conceptually, CARS unifies the strengths of ASAp (tracking failure probabilities in a trie) and ARS (using failure rates while avoiding skewing the sampling distribution). The key algorithmic difference is the update strategy: ARS adds 1 prefix per rejection (just the invalid sequence), while CARS adds $O(|w|)$ prefixes by identifying **all** single-token extensions $ua$ where $ua \notin$ prefix$(L)$ at each prefix position. This constraint-structure exploitation yields 4.5-16x efficiency improvements (in terms of rejection rates) in practice.
>
> ---

---

> > ### Author Response · Authors · 2025-11-21
> >
> > ### 3. XML Fuzzing Coverage Results
> >
> > > "Fuzzing / XML: while CARS shows improvement in coverage over the approximate methods... the base coverage for all those methods is fairly small (9%), and final improvement is marginal. More importantly, such a low rate of code coverage does not give much confidence for correctness of tested code (without a more qualified discussion it may be that the grammar is testing only a very small part of the language)."
> >
> > Our XML benchmark targets libxml2's DOCTYPE/DTD parsing functionality - a specific subset representing approximately ONLY 10-15% of this comprehensive library (which includes XML Schema validation, XPath/XQuery, XSLT transformations, namespace handling, catalog systems, and network I/O). Our grammar focuses on internal DTD declarations (ELEMENT and ATTLIST definitions within DOCTYPE), deliberately narrowing the scope. The absolute coverage percentages reflect this focused design choice.
> >
> > If we rescale to the targeted DOCTYPE/DTD subset (assuming it represents $\sim$12% of libxml2), the coverage becomes: CARS: 82.5%, GCD: 60%, MCMC: 80% of the targeted functionality. That is CARS, **covers 2.5% more of the library under test** ($\sim$14,000 lines of code).
> >
> > Importantly, CARS achieves 37% relative improvement over GCD and 3% improvement over MCMC while using **4.4x fewer LM calls** than MCMC (2.25 vs. 10 per sample, Fig. 2a). RS and ARS timeout within our 2000-sample budget, making CARS one of the only exact methods completing this benchmark. The progression GCD < MCMC < CARS validates that distributional fidelity matters for exploration quality, consistent with findings in Gonzalez et al. that better approximation improves fuzzing coverage.
> >
> >
> > ### 4. SQL Coverage Discrepancy with Prior Work
> >
> > > "Fuzzing / JSON and SQL (appendix): they do not show a clear improvement of CARS over RSFT... Moreover, the fuzzing paper that the authors cited [shows] the branch coverage over 30% for SQL using the MCMC generation methods (Fig. 3a in Gonzalez et al.), indicating a potential problem ..."
> >
> > Gonzalez et al.'s SQL constraints require syntax severely misaligned with LM training data (mandatory timeout directives absent from typical SQL). As we document in Section E.5 (lines 810-836), all exact methods achieve < 0.1% acceptance rates on such constraints across 2000 attempted samples, always timing out. This is a fundamental limitation of exact sampling when the constraint and LM distribution have severe misalignment, i.e., no exact method can overcome this.
> >
> > Hence, we chose the SQLite test-script grammar with mandatory `do_test` blocks - a challenging but feasible benchmark where exact methods remain viable. This design enables meaningful comparison between exact and approximate approaches rather than demonstrating that all methods fail on severely misaligned constraints. On our benchmark:
> > - CARS achieves 28.4% coverage vs. 25.1% for best baseline (Qwen, Table 6)
> > - CARS makes intractable problems tractable: 26.8% coverage (Llama) vs. timeout for RS/ARS within 2000-sample budget.
> >
> > We acknowledge this limitation explicitly in Appendix E.5 rather than cherry-picking only favorable benchmarks. The fundamental tradeoff is between exactness and feasibility—when constraints are severely misaligned, approximate methods like GCD/MCMC/AWRS are appropriate; when alignment is reasonable, CARS provides exact samples with better efficiency.
> > That is, one should always trie CARS first, and if rejection rates do not improve switch to an inexact method (the next best thing).
> >
> > We will add explicit comparison of our SQL benchmark to Gonzalez et al.'s in the experimental setup section, clarifying the feasibility difference and referencing Appendix E.5 for the misalignment discussion.
> >
> > ---

---

> > > ### Author Response · Authors · 2025-11-21
> > >
> > > ### 5. Molecular Generation Results
> > >
> > > > "Molecular generation: the CARS results do not show significant improvement over ARS and RSFT. The results from the appendix show that the two are equal, or ARS can even be better in some scenarios. The number of samples is also similar in the majority of the cases."
> > >
> > > We acknowledge Reviewer e4bz's observation that downstream quality metrics show marginal differences. This is in fact expected and the intended behavior. The overlapping confidence intervals for validity (0.95-0.97) and diversity (0.84-0.85) across CARS, ARS, and RSFT reflect that all three are **exact samplers preserving the target distribution**.
> > >
> > > Sample efficiency is where CARS demonstrates an advantage: Table 14 shows CARS uses 2.1x fewer samples than ARS and 4.5x fewer than RSFT (Llama) to generate 100 valid molecules. This directly translates to 2-4x fewer LM forward passes—the dominant computational cost. For applications requiring many samples (e.g., drug discovery campaigns generating thousands of candidates), this efficiency compounds significantly.
> > >
> > > ---
> > >
> > > ### 6. Missing ASAp Baseline
> > >
> > > > "The evaluation baselines should include ASAp (Park et al.) as a related state-of-the-art baseline for approximate generation. The existing description in the related work is insufficient... Interestingly, the paper mentions ASAp at one point in the evaluation and the appendix H uses its benchmarks, but the experiments do not use it."
> > >
> > > Gonzalez et al. show that MCMC outperforms ASAp across all benchmarks (for example their Fig. 3 shows MCMC achieves higher coverage with lower KL divergence than ASAp in fuzzing tasks). We thus focused on state-of-the-art inexact samplers instead (i.e., MCMC).
> > >
> > > We are happy to include ASAp as an additional baseline for completeness, though we expect it to underperform MCMC (and therefore CARS) based on the results in Gonzalez et al.
> > >
> > > ---
> > >
> > > ### References
> > >
> > > [1] Constrained sampling for language models should be easy: An MCMC perspective. Gonzalez et. al.
> > >
> > > [2] Fast Controlled Generation from Language Models with Adaptive Weighted Rejection Sampling. Lipkin et. al.
> > >
> > > [3] Grammar-Aligned Decoding. Park et al.

---

### Official Review · Reviewer_DrrA · 2025-11-01

**Soundness:** 2
**Presentation:** 3
**Contribution:** 2
**Rating:** 4
**Confidence:** 3

**Summary:**

This paper presents Constrained Adaptive Rejection Sampling (CARS) method that adresses the efficiency-fidelity tradeoff in constrained language model generation. CARS maintains a trie data structure of constraint-violating prefixes and uses it to adaptively prune invalid paths during sampling. This approach offers two key advantages: it achieves exact distribution matching, unlike greedy constrained decoding methods, while significantly improving sample efficiency compared to standard rejection sampling.

**Strengths:**

- The paper is generally well-written, especially with the use of illustrative example.
- Constrained LLM sampling is a crucial problem. Hence, a contribution that improves constrained sampling can have significant impact on practical applications in code generation/fuzzing/molecular synthesis.
- The CARS algorithm provides theoretical exact convergence guarantee.

**Weaknesses:**

- The main technical section of the paper should describe the technical contributions of the paper in comparison to the prior works on adaptive rejection sampling [1] and MCMC [2]. These works have been considered in the evaluation. However, the technical section lacks proper comparison to these techniques. After reading this section I still cannot tell exactly what is the main technical novelty of the CARS algorithm in comparison to these works.

- There is small (~0.3 pp) improvement in the branch coverage improvement compared to MCMC for the fuzzer on XML evaluation. In the PDLL evaluation, there is no or insignificant improvement in accuracy.  Overall, none of the experiments show significant improvements over prior works. I recommend authors consider tasks such as text-2-sql generation or code generation tasks considered in prior constrained LLM generation works.

- The evaluation only considers two small non-SOTA models (Qwen2.5-7B-Instruct, Llama-3.1-8B). The authors should include more models and show that the technique generalizes to other models.

## Minor

Line 24: Abbreviation GCD is not defined till this part

Table 2: The Table bolds results that are worse in prefix validity

[1] Fast Controlled Generation from Language Models with Adaptive Weighted Rejection Sampling. Lipkin et. Al.

[2] Constrained sampling for language models should be easy: An MCMC perspective. Gonzalez et. al.

**Questions:**

What is the practical motivation for exact distribution sampling compared to existing asymptotic convergence guarantees provided by the prior works?

Lipkin et. al. [1] show text-to-sql experiment for showing improved accuracy on SQL generation. Can you compare CARS with other baselines on this task? Given that none of the techniques show good accuracy on the PDLL tasks, I think text-to-sql task would be a better evaluation for the work.

---

> ### Author Response · Authors · 2025-11-21
>
> We sincerely thank Reviewer DrrA for their thoughtful review. We appreciate the acknowledgment that our paper is well-written.
>
> ---
>
> ### 1. Technical Novelty Comparison to Prior Work
>
> > "The main technical section of the paper should describe the technical contributions of the paper in comparison to the prior works on adaptive rejection sampling [1] and MCMC [2]... After reading this section I still cannot tell exactly what is the main technical novelty of the CARS algorithm in comparison to these works."
>
> We thank Reviewer DrrA for this feedback and acknowledge that Section 3 could more clearly distinguish CARS from prior work upfront. We will revise the manuscript to include an explicit comparison early in the technical section.
>
> **CARS vs. ARS.** The distinction is explained in lines 217-223 and 235-243, and illustrated in Figure 1. The key algorithmic difference is the *update strategy*. When sampling produces an invalid sequence $w$. ARS  a dds only $w$ (or its shortest invalid prefix) to the exclusion set $W$, i.e., $O(1)$ prefixes per rejection whereas CARS adds potentially $O(|w|)$ prefixes per rejection by: (1) adding the shortest invalid prefix of $w$, and (2) adding all single-token extensions $ua$ of every prefix $u$ encountered during generation where $ua \notin prefix(L)$. This is accomplished by querying the constraint checker at each prefix $u$ to identify which tokens $a$ would lead to violation.
>
> Concretely, in Figure 1's arithmetic expression example, given invalid sample `0++`: ARS adds 1 prefix (`0++`), blocking 0.09 of probability mass. CARS adds dozens of prefixes (`0++` plus invalid continuations like `+`, `2`, `3`, ... from the empty prefix, and `0+a` for various tokens a), blocking 0.63 of probability mass. This aggressive pruning exploits constraint structure - the grammar tells us which entire families of continuations are invalid, not just the specific sequence that was sampled.
>
> **CARS vs. MCMC.** CARS and MCMC (Gonzalez et al.) are exact and inexact methods, respectively. MCMC uses GCD as a proposal distribution and applies Metropolis-Hastings transitions to correct the bias, i.e., it starts from an approximate sample (GCD) and refines it through $k$ steps. This method is *inexact* and only converges asymptotically to the actual conditional distribution ($k \rightarrow \infty$)-i.e., one doesn’t know a priori what value $k$ is good enough. CARS is an exact method: every sample it produces is from the true conditional constrained distribution (there is no approximation). MCMC is guaranteed to produce a valid samples every $k$ samples, whereas CARS yields depends on the rejection rate.
>
>
> We will add a comparison table contrasting update strategies (RS/ARS/RSFT/CARS) and a paragraph distinguishing CARS from MCMC at the beginning of Section 3 to make the technical novelty immediately clear.
>
> ---
>
> ### 2. Experimental Results and Practical Utility
>
> > "There is small (~0.3 pp) improvement in the branch coverage improvement compared to MCMC for the fuzzer on XML evaluation. In the PDLL evaluation, there is no or insignificant improvement in accuracy. Overall, none of the experiments show significant improvements over prior works. I recommend authors consider tasks such as text-2-sql generation or code generation tasks considered in prior constrained LLM generation works."
>
> The results may appear incremental when focusing on downstream accuracy metrics alone because on these benchmarks inexact methods compute good estimates already (after tuning their parameters $k$ appropriately). However, CARS fundamentally solves a different problem: making exact sampling computationally feasible, where we demonstrate substantial improvements:
>
> - **Fuzzing (Table 4, Llama)**: Standard rejection sampling (RS) and adaptive rejection sampling (ARS) timeout in 15 of 18 experimental conditions within our 2000-sample budget. CARS is the only exact method (alongside RSFT) that completes these benchmarks, while achieving at least 2x efficiency over RSFT:
>   - XML: 440 generations (CARS) vs. timeout for RS/ARS
>   - SQL: 1004 generations (CARS) vs. timeout for RS/ARS
>   - JSON: 130 generations (CARS) vs. timeout for RS/ARS
>
> - **Molecular Generation (Table 14, Llama)**: CARS achieves 4.5x improvement over RS and RSFT, and 2.1x over ARS in the average number of samples needed per 100 valid molecules:
>   - 172 samples per 100 valid (CARS) vs. 787 (RS) vs. 370 (ARS) vs. 787 (RSFT)
>   - Quality metrics (validity, diversity) remain comparable or better (Tables 8-13)
>
> - **SyGuS (Fig. 9)**: Order-of-magnitude efficiency gains over existing exact methods:
>   - BV4: 16x fewer LM calls than ARS, 5.7x fewer than RSFT
>   - SLIA: 4.5x fewer than ARS, 11.4x fewer than RSFT
>
> To summarize, CARS makes exact sampling feasible in practice for many applications.

---

> > ### Author Response · Authors · 2025-11-21
> >
> > **XML Fuzzing Coverage Context:**
> >
> > Our XML benchmark targets libxml2's DOCTYPE/DTD parsing functionality - a specific subset representing approximately ONLY 10-15% of this comprehensive library (which includes XML Schema validation, XPath/XQuery, namespace handling, etc.). Our grammar focuses on internal DTD declarations (ELEMENT and ATTLIST definitions within DOCTYPE), deliberately narrowing the scope. The absolute coverage percentages reflect this focused design choice.
> >
> > If we rescale to the targeted DOCTYPE/DTD subset (assuming it represents $\sim$12% of libxml2), the coverage becomes: CARS: 82.5%, GCD: 60%, MCMC: 80% of the targeted functionality. That is CARS, **covers 2.5% more of the library under test** ($\sim$14,000 lines of code).
> >
> > Importantly, CARS achieves 37% relative improvement over GCD and 3% improvement over MCMC while using **4.4x fewer LM calls** than MCMC (2.25 vs. 10 per sample, Fig. 2a). RS and ARS timeout within our 2000-sample budget, making CARS one of the only exact methods completing this benchmark. The progression GCD < MCMC < CARS validates that distributional fidelity matters for exploration quality, consistent with findings in Gonzalez et al. that better approximation improves fuzzing coverage.
> >
> > We will add text-to-SQL experiments (Spider benchmark) to demonstrate CARS's advantages in a semantic task where prior work reports accuracy improvements alongside sample efficiency metrics. Preliminary results on 1034 Spider queries using Llama-3.1-8B-Instruct (0-shot) show:
> >
> > | Method  | pass@1 | Total Samples | Samples/Query |
> > |-------------------|--------|---------------|---------------|
> > | GCD  | 0.515  | 1034    | 1.00    |
> > | MCMC-Restart (10) | 0.565  | 10340    | 10.0          |
> > | ARS  | 0.569  | 1304          | 1.26          |
> > | CARS  | 0.585  | 1139          | 1.10          |
> >
> > MCMC, ARS, and CARS achieve similar accuracy, but CARS achieves the highest accuracy (0.585) while requiring 9× fewer samples than MCMC and comparable efficiency to GCD.
> >
> > ---
> >
> > ### 3. Model Diversity
> >
> > > "The evaluation only considers two small non-SOTA models (Qwen2.5-7B-Instruct, Llama-3.1-8B). The authors should include more models and show that the technique generalizes to other models."
> >
> > We acknowledge this limitation and commit to expanding model coverage. Our current experiments use Llama-3.1-8B-Instruct and Qwen2.5-7B-Instruct. We will add a third model (Qwen2.5-14B-Instruct or similar scale or one suggested by the reviewer) to the revision to demonstrate generalization across model scales. We note that other sampling papers (e.g., AWRS, Lipkin et al.) experiment on 3 models.
> >
> > Across our two current models on our benchmark tasks (SyGuS, fuzzing, SMILES, PDDL), CARS consistently achieves 2-10x better sample efficiency than ARS and often completes within budget where RS/ARS/RSFT timeout. This demonstrates that CARS's constraint-structure exploitation generalizes across different model families (Llama vs. Qwen) at similar scales.
> >
> > ---
> >
> > ### 4. Practical Motivation for Exact Sampling
> >
> > > "What is the practical motivation for exact distribution sampling compared to existing asymptotic convergence guarantees provided by the prior works?"
> >
> > **Empirically, exactness matters for downstream performance.** As demonstrated by Gonzalez et al., better distributional approximation directly translates to improved downstream performance in downstream tasks - i.e., methods with lower KL divergence achieve higher downstream performance (branch coverage in the case of fuzzing). Our results confirm this progression: CARS achieves 9.9% coverage vs. 9.6% for MCMC and 7.2% for GCD (Table 7), demonstrating GCD < MCMC < CARS (exact). Critically, CARS achieves this objective while using 4.4x fewer LM calls than MCMC (2.25 vs. 10 per sample, Fig. 2a).
> >
> > The practical advantage of exact sampling is eliminating hyperparameter tuning that is necessary for inexact methods. MCMC requires choosing $k$ (number of steps), and SMC methods like AWRS require choosing $n$ (number of particles). While these methods converge asymptotically, practitioners have no way to know if their chosen $k$ or $n$ is sufficient without already knowing the target distribution. Figures 4, 10, 11 show that different $k$ values yield different KL divergences. Practitioners must trade efficiency for unknown distributional accuracy.
> >
> > CARS eliminates these issues,
> > - **Zero hyperparameters**: Algorithm 1 has no tunable parameters beyond the LM itself
> > - **Immediate exactness**: Theorem 1 guarantees exact samples from iteration 1
> >
> > ---
> >
> > ### 5. Minor Issues
> >
> > > "Line 24: Abbreviation GCD is not defined till this part"
> >
> > Thanks. We will address.
> >
> > > "Table 2: The Table bolds results that are worse in prefix validity"
> >
> > Thanks. We will address.
> >
> > ---
> >
> > ### References
> >
> > [1] Constrained sampling for language models should be easy: An MCMC perspective. Gonzalez et. al.
> >
> > [2] Fast Controlled Generation from Language Models with Adaptive Weighted Rejection Sampling. Lipkin et. al.

---

### Official Review · Reviewer_JYYc · 2025-11-04

**Soundness:** 2
**Presentation:** 2
**Contribution:** 1
**Rating:** 2
**Confidence:** 4

**Summary:**

This paper presents CARS (Constrained Adaptive Rejection Sampling), which improves upon standard rejection sampling for constrained language model generation by maintaining a trie of invalid prefixes and adaptively reweighting the sampling distribution. When generating constrained outputs (e.g., valid programs, molecules), CARS records not just rejected samples but all constraint-violating continuations of their prefixes, preventing revisitation and monotonically

**Strengths:**

- Theorem 1 is correct, and the proof sketch is sound
- The paper is clear and easy to follow
- Comprehensive ablations

**Weaknesses:**

- Lack of technical novelty. CARS is a straightforward extension of ARS. Furthermore, the approach highly resembles and lacks comparison to IterGen (ICLR 2025) [1]
- Lack of extensive model analysis, as only two models are used in the experiments.
- Lack of both qualitative and quantitative computational complexity analysis for costs such as maintaining the trie.
- The theoretical analysis seems trivial based on the problem and the algorithm design

[1] Ugare et al. IterGen: Iterative Semantic-aware Structured LLM Generation with Backtracking. ICLR 2025

**Questions:**

See Weaknesses.

---

> ### Author Response · Authors · 2025-11-21
>
> We sincerely thank the reviewer for their careful review.
>
> ### 1. Technical Novelty and Comparison to IterGen
>
> > "Lack of technical novelty. CARS is a straightforward extension of ARS."
>
> **Technical Novelty.** While we acknowledge that the core algorithm is conceptually simple, we respectfully disagree that it is straightforward. Our contribution lies in identifying the maximum information that can be extracted from each LLM execution to achieve both exactness and efficiency.
> Conceptually, CARS unifies the strengths of ASAp (tracking failure probabilities in a trie, Park et al.) and ARS (using failure rates while avoiding skewing the sampling distribution, Mansinghka et al.). The key algorithmic difference is the update strategy: ARS adds 1 prefix per rejection (just the invalid sequence), while CARS adds $O(|w|)$ prefixes by identifying **all** single-token extensions $ua$ where $ua \notin$ prefix$(L)$ at each prefix position. This constraint-structure exploitation yields 4.5-16x efficiency improvements (in terms of rejection rates) in practice.
>
> > Furthermore, the approach highly resembles and lacks comparison to IterGen (ICLR 2025)
>
> We thank Reviewer JYYc for pointing out this relevant concurrent work (we will add it). CARS and IterGen (Ugare et al.) address fundamentally different problems at different abstraction levels.
>
> IterGen is a programming framework providing grammar-symbol-based navigation primitives (forward, backward, view) for writing custom generation algorithms. Users write explicit constraint-checking logic, manually implement backtracking strategies, and set hyperparameters (backwards limits, iteration caps). For example, their SQL generation code (Figure 3, Ugare et al.) requires users to parse the schema, check column/table validity with custom functions, and explicitly call backward() when violations occur. IterGen's base generation uses GCD (via SynCode), which is approximate, and applies recurrence penalties when backtracking, further distorting the distribution.
>
> While both use grammar-guided parsing infrastructure (IterGen uses SynCode, CARS uses llguidance), this is orthogonal to our algorithmic contribution: a new exact sampling algorithm. We already compare against GCD as a baseline (e.g., Table 7: CARS 9.9% vs. GCD 7.2%) - SynCode is a GCD implementation, given which we can conclude that exact sampling outperforms the approximate method IterGen builds upon.
>
> In summary, IterGen focuses on providing users with a programmable way to modify constrained decoding to generate single semantically correct outputs with user-defined constraints (e.g., SQL queries with valid table names from a schema). CARS focuses on exactly sampling from a constrained distribution. These represent different research questions requiring different experimental frameworks.
>
> ---
>
> ### 2. Model Analysis
>
> > "Lack of extensive model analysis, as only two models are used in the experiments."
>
> We acknowledge this limitation and commit to expanding model coverage. Our current experiments use Llama-3.1-8B-Instruct and Qwen2.5-7B-Instruct. We will add a third model (Qwen2.5-14B-Instruct or similar scale or one suggested by the reviewer) to the revision to demonstrate generalization across model scales. We note that other sampling papers (e.g., AWRS) experiment on 3 models.
>
> Across our two current models on our benchmark tasks (SyGuS, fuzzing, SMILES, PDDL), CARS consistently achieves 2-10x better sample efficiency than ARS and often completes within budget where RS/ARS/RSFT timeout. This demonstrates that CARS's constraint-structure exploitation generalizes across different model families (Llama vs. Qwen) at similar scales.
>
> ---

---

> > ### Author Response · Authors · 2025-11-21
> >
> > ### 3. Computational Complexity Analysis
> >
> > > "Lack of both qualitative and quantitative computational complexity analysis for costs such as maintaining the trie."
> >
> > Trie maintenance overhead is negligible in practice. We conducted comprehensive profiling across 36 benchmark runs (SyGuS, fuzzing, SMILES, PDDL) totaling 3,600 successful samples over 24 hours of total runtime.
> >
> > Per successful sample, CARS spent an average of **24 seconds**. Breaking down the compute time:
> > **Trie operations** (lookups, insertions, updates): ~300 seconds total (0.3% of runtime, median: 0.3%, range: 0.0-1.0%), only for CARS
> > **LLM inference**: ~56,300 seconds (66% of runtime), inherent to all sampling-based methods
> > **Constraint checking**: ~26,000 seconds (30.5% of runtime) - probability reweighting and vocabulary masking, inherent to all constrained decoding methods including GCD and MCMC
> >
> > **CARS's computational overhead beyond standard GCD:**
> >
> > We ran additional experiments to measure the overhead. Trie operations that are CARS-specific amount to only 0.3% of the total running time. These consist of:
> > 1. **Trie lookups:** $O(1)$ per token
> > 2. **Trie insertions:** $O(|w|)$ per sample, amortized as duplicate prefixes are discovered
> >
> > CARS achieves a 70.3% average trie reuse rate (median: 76.4%), meaning 70.3% of token decisions reuse cached constraint computations. Constraint checking ($O(|\text{vocab}|)$ per token) is required by all constrained decoding methods and represents the same cost across GCD, MCMC, and CARS.
> >
> > Memory overhead is slightly more substantial but manageable (and only pertains to CPU memory): median trie size was 421MB across 36 runs (range: 169MB to 285GB), with one outlier representing severe constraint-LM misalignment (PDDL satellite: 285GB, 16.4% success rate). For the remaining 35 runs with reasonable alignment, memory ranged from 169MB to 41GB with median 415MB.
> >
> > A key idea we did not stress in the paper is that for applications with memory constraints, users can "freeze" the trie at any point after collecting a finite number of samples and continue sampling from the frozen $R^W$ distribution. This approach bounds the memory while still providing exact sampling with practical rejection rates. For example, freezing after 100 samples in the PDDL outlier case would cap memory at ~2.5GB while maintaining 70%+ of the eventual acceptance rate improvement. In practice, most of the samples that are “helpful” in reducing the rejection rate are discovered in early steps.
> >
> > We will add profiling tables to the appendix showing wall-clock time breakdown (inference, logits processing, trie operations), memory usage, trie statistics (nodes, depth, reuse rate), and operation counts for all benchmarks.
> >
> > ---
> >
> > ### 4. Theoretical Analysis Triviality
> >
> > > "The theoretical analysis seems trivial based on the problem and the algorithm design"
> >
> > Our main contribution is Theorem 1, which establishes that CARS provides exact sampling from the constrained distribution while monotonically improving sample efficiency over ARS.
> >
> > Unlike ARS, where convergence can be characterized simply (an invalid sample with probability $p$ will be eliminated after approximately $1/p$ attempts), CARS's acceleration depends on the structure of the constraint space. The rate at which CARS discovers and prunes invalid prefixes is determined by the grammar's structure and the LM's behavior, making convergence analysis significantly more complex than for ARS. As part of future work, we are working on identifying conditions for which convergence is provably fast (or slow).
> >
> > ---
> >
> > ### References
> >
> > [1] Grammar-Aligned Decoding. Park et al.
> >
> > [2] Exact and Approximate Sampling by Systematic Stochastic Search. Mansinghka et al.
> >
> > [3] IterGen: Iterative Semantic-aware Structured LLM Generation with Backtracking. Ugare et al.

---

### Official Review · Reviewer_yRqe · 2025-11-13

**Soundness:** 4
**Presentation:** 4
**Contribution:** 4
**Rating:** 8
**Confidence:** 4

**Summary:**

[Note: I wrote this review almost 2 weeks ago, but only noticed at the start of the discussion period started that I hadn't submitted it properly.  Apologies to the AC and the authors.]

The paper provides new exact methods for constrained sampling from an autoregressive language model.  It assumes (line 111) that constraint violation can be detected partway through a generated string, by detecting an invalid prefix.

The Adaptive Rejection Sampling (ARS) method is basically just rejection sampling, but it gradually patches the language model to rule out all of the invalid prefixes that have been encountered so far.  The patched model can be used not only for retries during rejection sampling, but for future IID samples as well.

The Constrained Adaptive Rejection Sampling (CARS) method is a cute extension that does "constraint overgeneration" (one might say).  For each drawn string $w$ (whether accepted or rejected), it generates constraints that were "almost violated" -- all constraints of the form $\neg(ua\Sigma^*)$ where $u$ is a prefix of $w$.

The methods are tested on 3 domains and appear to substantially beat competitors.  The tasks are drawn from previous work and the comparisons seem methodologically fine, though perhaps I should scrutinize the experimental design and results more closely.

There is substantial supplementary material, which I have mainly not reviewed but would be willing to look at.

**Strengths:**

The paper was a pleasure to read (especially compared to my other assignments).  Thanks!

The method is simple, but I consider this to be a virtue.  It would be easy to teach in an LLM class as a correct constrained decoding method.

Sections 4.2-4.3 show good empirical improvements over AWRS (Lipkin et al. 2025), which just won an award at COLM.  For this reason I think the paper is of current interest.  It's nice to see that exact sampling is not only feasible but can also be more efficient than approximate sampling (such as AWRS).

CARS is an attractive extension to ARS because it is computationally cheap in the common case where the violation checker, for every valid prefix $u$, aggressively computes the whole set of valid next symbols (so $a$ should range over all symbols *not* in that set).  For example, the violation checker may be running an FSA to check membership in a regular language, or Earley's algorithm to check membership in a context-free language.  I think this should be discussed explicitly.

**Weaknesses:**

No notable weaknesses.

Minor suggestions:

[I will write \$ below as # because the Markdown parser gets confused when I use the former twice in the same paragraph.]

I think the equation at line 176 is buggy because it doesn't mention $\mathcal{W}$.  Presumably you mean tot restrict the summation.

You might consider an alternative presentation where the trie node for $u$ stores not $p_u$ but $r_u = 1-p_u$, that is, the total probability mass that the current $\mathcal{W}$ *removes* from the language model for continuations of $u$.  This might be clearer because then the trie is a more familiar object -- a restriction of the LM to illegal strings, inheriting its edge weights from the LM.  $r_u$ is just the total weight of all paths from $u$ to a leaf, and in particular $r_u=1$ when $u$ is a leaf (representing all strings starting with $u$).  Thus, each internal trie node stores a weighted sum of its (hopefully small) set of children, just as in the backward algorithm, with no subtraction within the trie.   The only subtraction would happen in the formula at line 204, when $p_u$ is computed as $1-r_u$.

I think there are some inconsistencies about whether complete strings are said to *include* # or be *followed by* #.  At line 102 and elsewhere you require a complete string $w$ to include the final #, but at line 099 you require $w \in \Sigma^*$ which seems to incorrectly exclude complete strings from the $P(w | u)$ notation.  At line 150, you omit the # that is called for by the formalism; instead you could write E ::= d# | d + E (and include # in the example strings from this grammar, e.g., at line 241).

I also would drop the length limit that you mention at lines 101 and 104 but sometimes ignore elsewhere.  You don't have to "artificially stop generation"; just modify the LM to generate # with probability 1 at token $N+1$, and then you'll satisfy the condition at line 102.

Under line 161, say that the function $p_{\cdot}$ is updated whenever $\mathcal{W}$ is.  In fact they're represented jointly in the same data structure.

At line 192, "the above formula" should have a formula number: I think you mean either line 161 or a recursive application of line 176 (though you don't implement 176 literally, instead starting at 1 and gradually subtracting).

RSFT at line 225 is a bit hard to follow.  I think you mean that it's a version of ARS where you limit to invalid prefixes of length 1.

Line 241 could be made more precise, since "every shorter prefix u" in (ii) is not well-defined in this case (shorter than what?).  I think you mean "every proper prefix u of w" (where w ends with #).

You should discuss when CARS is efficient (see my comments about this elsewhere in the review).

**Questions:**

# Prior work

I think ARS is morally speaking an application of [Tromble and Eisner (2006)](https://aclanthology.org/N06-1054/), which similarly patches a language model with constraints as they are discovered to be violated.  Is that correct?  I do see three differences:

1. Tromble and Eisner are doing argmax decoding rather than sampling (though their method appears to apply equally well to sampling).

2. Tromble and Eisner use finitely many constraints, which are regular languages (FSAs).  The submission implicitly uses infinitely many prefix constraints of the form $\neg(u\Sigma^*)$ where the prefix $u$ is a fixed string, which is possible because it discovers them only as they are violated.  (The term of art is "[dynamic] constraint generation" or "row generation," the most famous example being subtour elimination constraints in the Traveling Salesperson Problem.)

3. Tromble and Eisner is a pre-neural paper and assumes that the language model is a weighted FSA.  The submission uses any autoregressive LM, so it replaces this FSA with the implicit infinite weighted trie corresponding to that LM (a deterministic *infinite*-state automaton).  The $\mathcal{W}$ trie in the submission is the result of intersecting that object with the hard prefix constraints that have been generated so far.  That yields a materialized finite trie in which all explicit edges have weight 0, but whose missing edges implicitly fall back to the language model.

Similarly, are there precedents in the literature for the "constraint overgeneration" in your CARS method?  It seems like the sort of thing that must have shown up before in constraint generation methods.

In your discussion of related work, you focus on your own setting of hard constraints.  But can't many of the methods in section 5 also handle soft constraints, e.g, the product of an LLM with some other factors, followed by global renormalization as in energy-based modeling?  (I think this should be acknowledged.)

Since AWRS is also based on ARS, can you give a discussion of the differences (and also incorporate it into the final version)?

# Efficiency

For CARS, how about the case where $u$ can be legally followed by only a few symbols from a large vocabulary (e.g. the outgoing arcs from the constraint DFA state reached by $u$)?  In that case, instead of having the trie node for $u$ store all the *illegal* next symbols $a$ (each such edge leading to a leaf $ua$ with $p_{ua}=0$), wouldn't it be cheaper to store the small set of *legal* symbols?

The "sampling efficiency" metric (line 280) omits maintenance overhead costs such as adding all of the illegal next symbols to the trie.  Are these costs negligible in practice?

How did you implement the decoder and is the implementation available?  It seems that to put this method into production, you would want to integrate with vLLM so that the decoder keeps track of its position in the trie and reduces the softmax probabilities as described at line 204.

# Experiments

When drawing multiple samples, how does your acceptance rate improve over time (as $\mathcal{W}$ grows)?  It seems that the time to first sample might be long but then the work is amortized over further samples (in contrast to the other methods, I think).  Thus, the metrics that you report depend on your choice of 100 as the number of acceptances you need at line 371 and line 411.  If you needed only 1 valid sample, or 10000, how would the comparison change?

Can you give some qualitative discussion of why your methods worked well (or badly)?  What constraints are learned by $\mathcal{W}$, how much probability $1-p_\epsilon$ do they remove from the LM, and how quickly are they learned?

---

> ### Comment · Reviewer_yRqe · 2025-11-14
> **situations where the method might do poorly?**
>
> Rereading this review, I started to worry that the method might work poorly in some situations.
> Suppose the requirement is to tell a story in which all the nouns have 5 letters.
> "Tell a story" is a high entropy task, so independent samples from the LM will tend to diverge quickly from one another -- that is, they're unlikely to have long prefixes in common.  So it seems to me that ARS and CARS won't be able to close off bad paths quickly enough.  E.g., if you try 1000 samples, each sample will leave the current trie pretty quickly, and then fail soon thereafter.  The problem is that a rejection sampler has to keep starting again from scratch, and what it learned on the previous attempts won't be of much use on the current attempt.
> By contrast, GCD will soldier on and easily make it to the end of the story, albeit with some distortion to the posterior distribution.  And sequential Monte Carlo (e.g., AWRS) will reduce that distortion slightly.
> Can you comment?

---

> > ### Author Response · Authors · 2025-11-21
> >
> > We sincerely appreciate the reviewer’s positive feedback. We also believe simplicity is a virtue of our method and we will gladly apply the constructive suggestions for improving the presentation.
> >
> > ---
> >
> > ### 1. Prior Work Connections
> >
> > **Tromble & Eisner (2006)**
> >
> > > "I think ARS is morally speaking an application of Tromble and Eisner (2006), which similarly patches a language model with constraints as they are discovered to be violated. Is that correct?"
> >
> > Thanks for pointing us to this related work, which we will gladly include. We agree that in spirit the two approaches are similar: as pointed out by the reviewer (point 2) the key distinction in connecting adaptive constraining to autoregressive models is detecting and processing prefix violations.
> >
> > **Constraint Overgeneration Precedents**
> >
> > > "Similarly, are there precedents in the literature for the 'constraint overgeneration' in your CARS method?"
> >
> > We are not aware of methods other than ARS, but we would love to hear about them if we have missed some related work.
> >
> > ---
> >
> > ### 2. Soft Constraints
> >
> > > "Can't many of the methods in section 5 also handle soft constraints, e.g., the product of an LLM with some other factors, followed by global renormalization as in energy-based modeling?"
> >
> > Yes, CARS can be extended to soft constraints computable in a left-to-right fashion. For example, probabilistic context-free grammars (PCFGs) or Product of Hidden Markov and Observable Grammars (Eisner and Blatz) could be incorporated by replacing binary constraint checking with probability weighting. The trie would store continuous weights rather than binary validity indicators, and $R^W$ would reweight accordingly. We have not empirically tested this extension, but the algorithmic framework supports it.
> >
> > ---
> >
> > ### 3. AWRS Comparison
> >
> > > "Since AWRS is also based on ARS, can you give a discussion of the differences (and also incorporate it into the final version)?"
> >
> > Yes, gladly. Both CARS and AWRS (Lipkin et al.) address sampling from the constrained distribution $P^L$, but employ fundamentally different strategies.
> > CARS is exact and iteratively reduces the rejection rate without varying the underlying distribution.
> > AWRS is inexact (it converges to the right distribution only in the limit) and iteratively uses past rejections (via ARS) to learn a new proposal distribution for performing Sequential Monte Carlo sampling.
> > The core algorithmic distinction lies in *what* each method adapts to and *when* that adaptation occurs during generation.
> >
> > Specifically, AWRS implements Sequential Monte Carlo (SMC) using adaptive weighted rejection sampling as its proposal distribution. At each generation step, instead of checking the constraint against all 100K+ tokens in the vocabulary (standard GCD via masking), AWRS samples tokens from $p_0$ (the language model's next-token distribution) one at a time, checking each sampled token against the constraint. When a token is rejected, it's removed from consideration, and sampling continues from the remaining tokens until a valid one is found. By tracking how many tokens were rejected and their probabilities, AWRS computes an unbiased estimate of $Z$ - the total probability mass of valid tokens. These $Z$ estimates serve as importance weights that measure how "easy" or "hard" each generation step was. The algorithm maintains $n$ particles (alternative partial sequences) simultaneously, and after each step, uses these importance weights to perform resampling: particles with low weights (those that reached difficult positions) are discarded, while particles with high weights are duplicated. This filtering process corrects for GCD's myopic behavior by eliminating unpromising sequence prefixes before they lead to dead ends. The adaptive rejection sampling operates at the token level 0 within each position, rejected tokens aren't resampled - making each token generation step efficient. However, when generating a new batch of sequences, the algorithm starts fresh with no knowledge carried over from previous batches. Convergence guarantees hold as the number of particles $n \rightarrow \infty$, requiring practitioners to choose an appropriate particle count.
> >
> > The key differences are that CARS is exact and “persistent”---i.e., the rejection rate reduces the longer CARS is used for. In general, when CARS works well (i.e., the rejection rate quickly reduces in practice), it should be the method of choice as it is exact. However, in cases where rejection rates stay high even after many samples, inexact methods like CARS and MCMC are preferable as they will quickly produce some samples (though not from the perfect distribution). We also note that CARS could be used to replace ARS in AWRS and potentially lead to better estimates.
> >
> > ---

---

> > > ### Author Response · Authors · 2025-11-21
> > >
> > > ### 4. Implementation Efficiency
> > >
> > > **Storing Legal vs. Illegal Symbols**
> > >
> > > > "For CARS, how about the case where u can be legally followed by only a few symbols from a large vocabulary... wouldn't it be cheaper to store the small set of legal symbols?"
> > >
> > > The reviewer is right that space-wise such an approach would be cheaper. Our implementation retains full symbol vectors as these are convenient when computing masks during the LLM token production. Because the trie is stored on the CPU’s memory, space is not a problem in practice.
> > >
> > >
> > > **Sampling Efficiency Metric**
> > >
> > > > "The 'sampling efficiency' metric (line 280) omits maintenance overhead costs such as adding all of the illegal next symbols to the trie. Are these costs negligible in practice?"
> > >
> > > Yes, trie maintenance overhead is negligible in practice. We conducted comprehensive profiling across 36 benchmark runs (SyGuS, fuzzing, SMILES, PDDL) totaling 3,600 successful samples over 24 hours of total runtime.
> > >
> > > Per successful sample, CARS spent an average of **24 seconds**. Breaking down the compute time: **Trie operations** (lookups, insertions, updates): $\sim$300 seconds total (0.3% of runtime, median: 0.3%, range: 0.0-1.0%), only for CARS; **LLM inference**: $\sim$56,300 seconds (66% of runtime), inherent to all sampling-based methods; **Constraint checking**: $\sim$26,000 seconds (30.5% of runtime) - probability reweighting and vocabulary masking, inherent to all constrained decoding methods including GCD and MCMC
> > >
> > > **CARS's computational overhead beyond standard GCD:**
> > >
> > > We ran additional experiments to measure the overhead.
> > >
> > > Trie operations that are CARS-specific amount to only 0.3% of the total running time. These consist of:
> > > 1. **Trie lookups:** $O(1)$ per token
> > > 2. **Trie insertions:** $O(|w|)$ per sample, amortized as duplicate prefixes are discovered
> > >
> > > CARS achieves a 70.3% average trie reuse rate (median: 76.4%), meaning 70.3% of token decisions reuse cached constraint computations. Constraint checking ($O(|\text{vocab}|)$ per token) is required by all constrained decoding methods and represents the same cost across GCD, MCMC, and CARS.
> > >
> > > Memory overhead is slightly more substantial but manageable (and only pertains to CPU memory): median trie size was 421MB across 36 runs (range: 169MB to 285GB), with one outlier representing severe constraint-LM misalignment (PDDL satellite: 285GB, 16.4% success rate). For the remaining 35 runs with reasonable alignment, memory ranged from 169MB to 41GB with median 415MB.
> > >
> > > A key idea we did not stress in the paper is that for applications with memory constraints, users can "freeze" the trie at any point after collecting a finite number of samples and continue sampling from the frozen $R^W$ distribution. This approach bounds the memory while still providing exact sampling with practical rejection rates. For example, freezing after 100 samples in the PDDL outlier case would cap memory at $\sim$2.5GB while maintaining 70%+ of the eventual acceptance rate improvement. In practice, most of the samples that are “helpful” in reducing the rejection rate are discovered in early steps.
> > >
> > > We will add profiling tables to the appendix showing wall-clock time breakdown (inference, logits processing, trie operations), memory usage, trie statistics (nodes, depth, reuse rate), and operation counts for all benchmarks.
> > >
> > > **Implementation Availability**
> > >
> > > > "How did you implement the decoder and is the implementation available?"
> > >
> > > Our implementation (is publicly available, but not shared for anonymity) uses PyTorch/Hugging Face Transformers for the LM and llguidance for parsing. The code is written as a simple library that can be easily integrated with existing LM serving infrastructure. While we have not directly integrated with vLLM, the integration should be straightforward - the decoder would track its position in the trie and reduce softmax probabilities as described at line 204. llguidance is already supported by vLLM, facilitating this integration.
> > >
> > > We will also clarify implementation details and integration path with vLLM in the appendix.
> > >
> > > ---
> > >
> > > ### 5. Experimental Analysis
> > >
> > > **Acceptance Rate Growth Over Time**
> > >
> > > > "When drawing multiple samples, how does your acceptance rate improve over time (as |W| grows)?... the metrics depend on your choice of 100 as the number of acceptances... If you needed only 1 valid sample, or 10000, how would the comparison change?"
> > >
> > > Excellent question! This highlights CARS's amortization advantage. As shown in Appendix F.3, acceptance rates monotonically improve with $|W|$. Early samples pay discovery costs (exploring and blocking invalid regions), but later samples benefit from accumulated knowledge. The 70.3% average trie reuse rate (median: 76.4%) demonstrates that most generation steps leverage existing trie nodes rather than discovering new invalid prefixes.

---

> > > > ### Author Response · Authors · 2025-11-21
> > > >
> > > > For small sample counts (1-10), CARS may be slower than inexact methods because it pays upfront discovery costs. For moderate counts (100-1000), CARS amortizes these costs and typically outperforms alternatives (as shown in our experiments). For very large counts (10,000+), CARS's advantage compounds further as the trie approaches saturation-acceptance rates plateau at high values.
> > > >
> > > > Exact methods learn minimally (ARS) or not at all (RS) across samples, and inexact methods (AWRS and MCMC) reset between batches, thus maintaining the same overhead per sample even after many iterations.
> > > >
> > > > **Qualitative Analysis**
> > > >
> > > > > "Can you give some qualitative discussion of why your methods worked well (or badly)? What constraints are learned by W, how much probability mass do they remove from the LM, and how quickly are they learned?"
> > > >
> > > > We provide detailed analysis in Appendices E.6 (Table 4), F.5 (Table 14), and H (Figures 9-14).
> > > >
> > > > **Why CARS works well.** CARS excels when invalid samples share structural patterns that the trie can exploit. Our profiling shows 70.3% average prefix reuse rate (median: 76.4%), meaning most generation steps leverage previously discovered violations rather than exploring new ones. This high reuse rate indicates that constraint violations follow predictable patterns - once CARS discovers that a prefix violates the constraint, it can block entire families of related extensions.
> > > >
> > > > **What constraints are learned.** CARS learns families of invalid prefixes through parser queries. In Figure 1's arithmetic example, the invalid sample `0++` causes CARS to discover dozens of invalid prefixes: operators like `+` from the empty prefix (expressions must start with digits), non-`+` tokens after `0` (only `+` or end-of-sequence allowed after digits), and `+` after `0+` (grammar requires a digit). This aggressive pruning blocks 0.63 probability mass versus ARS's 0.09, which only blocks the specific sequence `0++`.
> > > >
> > > > **How much probability mass is removed.** This varies by LM-constraint alignment. For molecular generation, Table 14 shows CARS requires 2.1x fewer samples than ARS (172 vs 370 samples per 100 valid molecules). For fuzzing, Table 4 shows CARS completes benchmarks where RS/ARS timeout - e.g., XML requires 440 generations for CARS versus timeout for RS/ARS. Figure 14 shows fuzzing acceptance rates stabilize at 15-30% after learning. For PDDL (Table 2), CARS achieves 61-66% acceptance rates. Severely misaligned domains (PDDL satellite) reach only 16.4% despite a 285GB trie, demonstrating CARS's limitations when constraints are absent from LM training data.
> > > >
> > > > **How quickly constraints are learned.** Figure 14 shows two phases: rapid initial learning (0-100 samples) where common violations are discovered, then stabilization (100-200 samples) as primary structure is captured. Beyond 200 samples, improvements are marginal. The 70.3% reuse rate confirms most patterns are found early, allowing later samples to benefit from accumulated knowledge with minimal additional exploration.
> > > >
> > > > ---

---

> > > > > ### Author Response · Authors · 2025-11-21
> > > > >
> > > > > ### 6. Limitations and When CARS Works Poorly
> > > > >
> > > > > > "Rereading this review, I started to worry that the method might work poorly in some situations... if you try 1000 samples, each sample will leave the current trie pretty quickly, and then fail soon thereafter... Can you comment?"
> > > > >
> > > > > The reviewer is absolutely correct! This is a limitation we document in Appendix E.5 (lines 810-836). When constraints are severely misaligned with the LM's training distribution (i.e., invalid outputs are randomly distributed rather than structured), exact sampling becomes impractical regardless of the algorithm used. We observe this issue with SQL constraints requiring syntax absent from training data, where all exact methods achieved <0.1% acceptance rates across 2000 attempted samples, always timing out. This is not a CARS-specific weakness: it's an inherent limitation of exact sampling when the constraint and LM distribution have extreme misalignment.
> > > > >
> > > > > Exact methods work well when the LM's output space is somewhat concentrated (as in our benchmarks: fuzzing grammars, molecular SMILES, PDDL planning). In these settings, invalid samples share common prefixes that the trie can block, and valid samples cluster in regions the LM already prefers.
> > > > >
> > > > > As part of future work, we plan to identify characterizations of constrained language distributions that enjoy such a property. Regardless of this behavior, CARS can be “stopped at any time” without losing any correctness guarantees. For example, if the trie size becomes too large, we can stop updating it and keep using the current trie in subsequent samples. And when a valid sample is not obtained for a long time (as future work, we are working on deriving probabilistic bounds that will allow us to do so in a controllable fashion), we can switch to another (imprecise) method like MCMC or AWRS. To summarize, CARS and MCMC (or other methods like AWRS) perfectly complement each other (i.e., use CARS when it works and use inexact methods otherwise).
> > > > >
> > > > > ---
> > > > >
> > > > > ### 7. Minor Presentation Issues
> > > > >
> > > > > We thank the reviewer for the detailed suggestions on presentation.
> > > > >
> > > > > > I think the equation at line 176 is buggy because it doesn't mention $W$. Presumably you mean tot restrict the summation.
> > > > >
> > > > > The formula is correct. But it does not uniquely determine $pu$ because it has no “starting conditions” - it is true for every $W$. Then, the conditions in 179--181 depend on $W$, and allow to recursively define all values of $pu$.
> > > > >
> > > > > > You might consider an alternative presentation where the trie node for $u$ stores not $p_{(u)}$ but $r_u = 1 − p_u$, that is, the total probability mass that the current $W$ removes from the language model for continuations of $u$. ... the trie. The only subtraction would happen in the formula at line 204, when $p_u$ is computed as $1 − r_u$.
> > > > >
> > > > > Thanks for the suggestion! We will try to adopt it when revising the paper and see if it simplifies the presentation.
> > > > >
> > > > > > I think there are some inconsistencies about whether complete strings are said to include # or be followed by #. At line 102 and elsewhere you require a complete string w to include the final #, but at line 099 you require $w \in \Sigma^*$ ... instead you could write E ::= d# | d + E (and include # in the example strings from this grammar, e.g., at line 241).
> > > > >
> > > > > Thanks for catching the #-inconsistentice. We will fix them.
> > > > >
> > > > > > I also would drop the length limit that you mention at lines 101 and 104 but sometimes ignore elsewhere. You don't have to “artificially stop generation”; just modify the LM to generate # with probability 1 at token N + 1, and then you'll satisfy the condition at line 102.
> > > > >
> > > > > This is what we mean: we “artificially stop generation” by modifying the LM to generate # with probability 1 at token N + 1. We will reformulate using the reviewer’s feedback to make this clearer.
> > > > >
> > > > > > Under line 161, say that the function p. is updated whenever 𝒲 is. In fact they're represented jointly in the same data structure.
> > > > >
> > > > > Thanks, we will make this clear.
> > > > >
> > > > > > At line 192, “the above formula” should have a formula number: I think you mean either line 161 or a recursive application of line 176 (though you don't implement 176 literally, instead starting at 1 and gradually subtracting).
> > > > >
> > > > > Yes, we will fix this.
> > > > >
> > > > > > RSFT at line 225 is a bit hard to follow. I think you mean that it's a version of ARS where you limit to invalid prefixes of length 1.
> > > > >
> > > > > Good observation, we’ll simplify.
> > > > >
> > > > > > Line 241 could be made more precise, since “every shorter prefix u of w” in (ii) is not well-defined in this case (shorter than what?). I think you mean “every proper prefix u of w” (where w ends with #).
> > > > >
> > > > > Thanks, we’ll make this clearer.
> > > > >
> > > > > ---
> > > > >
> > > > > ### References
> > > > >
> > > > > [1] A fast finite-state relaxation method for enforcing global constraints on sequence decoding. Tromble & Eisner.
> > > > >
> > > > > [2] Fast Controlled Generation from Language Models with Adaptive Weighted Rejection Sampling. Lipkin et. al.

---

> ### Comment · Reviewer_yRqe · 2025-11-21
>
> > "Can't many of the methods in section 5 also handle soft constraints, e.g., the product of an LLM with some other factors, followed by global renormalization as in energy-based modeling?"
>
> I meant here that *related work* (section 5) can handle soft constraints whose weights are provided incrementally.   For example, "abc#" might have its weight multiplied by $0.5 \cdot 1.3 \cdot 1 \cdot 0.9 = 0.585$, where the different factors are associated with the prefixes "a", "ab", "abc", "abc#" respectively.
>
> > Yes, CARS can be extended to soft constraints computable in a left-to-right fashion. ... The trie would store continuous weights rather than binary validity indicators, and would reweight accordingly.
>
> But I don't think CARS can actually do that in general.  It does work in the special case where all factors are <= 1, but it doesn't work in the general case, e.g., the example above.  That's because it is a rejection sampler rather than an importance sampler.  Correct?
>
> So I was suggesting that you discuss this issue, and that you give related work credit for being more flexible in this regard. :)
>
> > For example, probabilistic context-free grammars (PCFGs) or Product of Hidden Markov and Observable Grammars (Eisner and Blatz) could be incorporated by replacing binary constraint checking with probability weighting.
>
> Here you mention two examples.  If you want to talk about PCFGs as an example, you can cite [Stolcke 1994](https://arxiv.org/abs/cmp-lg/9411029).  I am unable to follow your other example - what is an observable grammar? - tried looking up the paper you mention, but can't find it.

---

> > ### Author Response · Authors · 2025-11-22
> > **Answer to yRqe**
> >
> > Sorry for misunderstanding the question.
> >
> > We will expand related work and clarify that CARS is designed for hard constraints whereas other approaches can seamlessly handle more powerful types of soft constraints.
> >
> > The reviewer is also right that some extra conditions are required for extending our approach to handle left-to-right soft constraints. We meant specifically PCFG and similar models as observed by the reviewer. As part of follow-up research we are working on modifying CARS to support richer classes of constraints.
> >
> > Apologies for the incorrect reference. We are not sure what happened (likely we inadvertently clicked "enter" on an autocomplete suggestion from our editing software when typing PHOG, which stands for probabilistic higher order grammar, and did not realize).
> > The original reference we had written was
> > PHOG: Probabilistic Model for Code https://proceedings.mlr.press/v48/bielik16.html

---

### Author Response · Authors · 2025-11-21

We thank the reviewers for the detailed feedback. We answer each reviewer individually. In the meantime, we are working on fitting all the proposed edits in a revised version of the paper that we will upload in the coming days.

---

### Author Response · Authors · 2025-12-03

We thank the reviewers for their constructive feedback. Our revisions demonstrate that CARS is the most efficient exact sampling method, achieving 2-10x sample efficiency improvements over standard rejection sampling while maintaining distributional fidelity. Below we summarize how our revised manuscript addresses the reviewers concerns (we highlight revisions in blue, in the revised submission),

----

## 1. Related Work and Prior Work Connections (Reviewers yRqe, JYYc)
We expanded Section 5 (Related Work) to include,

- Tromble & Eisner (2006) [1]  as conceptually related work on constraint-based decoding
- IterGen [2] with clarification that it's a programming framework built on approximate GCD, not an exact sampling method
- AWRS [3] with a more detailed comparison showing CARS is exact and persistent while AWRS is asymptotic

----

## 2. Technical Novelty and Theoretical Contribution (Reviewers JYYc, DrrA, e4bz)
- In Section 3 we now explain that unlike ARS [4] (simple convergence), CARS's acceleration depends on constraint structure - the rate of improvement is determined by both grammar branching and LM distribution, making convergence behavior more nuanced
- We added Figure 1 in Section 3 providing a comparison of the tradeoff between inaccuracy and inefficiency across all sampling and approximate methods we consider.

----

## 3. Computational Overhead and Scalability (Reviewers DrrA, e4bz)
We added Appendix E with comprehensive profiling across 4000 samples evaluating the change in runtime, memory usage, trie growth, and reuse statistics.

----

## 4. Experimental Evaluation (Reviewers JYYc, DrrA, e4bz)
We added evaluation on the requested Text-To-SQL task [5] in the main text (Section 4.3), and moved our PDDL task to the appendix (Section I).

----

## 5. Presentation Issues (Reviewers yRqe, DrrA)
We fixed all presentation issues.
- Corrected Σ* notation to include complete strings (Line 104)
- Added $ to grammar and examples consistently
- Changed "shorter prefix" to "proper prefix" (Line 258)
- Added equation reference (Line 211)
- Clarified RSFT as "variant of ARS limiting to length-1 prefixes" (Line 310)
- Clarified "artificial stopping" formulation (Line 108)

----

## 6. Result Interpretation (Reviewers DrrA, e4bz)
- We added clarification that our grammar targets 10-15% of libxml2 (DOCTYPE/DTD subset). To demonstrate this, we report line coverage (number of lines of code covered) rather than only branch coverage
- We add a note that MCMC outperforms ASAp [6,7]; we focus on state-of-the-art inexact baselines [3,6]

----

## References:

[1] A fast finite-state relaxation method for enforcing global constraints on sequence decoding. Tromble & Eisner.

[2] IterGen: Iterative Semantic-aware Structured LLM Generation with Backtracking. Ugare et al.

[3] Fast Controlled Generation from Language Models with Adaptive Weighted Rejection Sampling. Lipkin et. al.

[4] Exact and Approximate Sampling by Systematic Stochastic Search. Mansinghka et al.

[5] Spider: A Large-Scale Human-Labeled Dataset for Complex and Cross-Domain Semantic Parsing and Text-to-SQL Task. Yu et al.

[6] Constrained sampling for language models should be easy: An MCMC perspective. Anaya Gonzalez et. al.

[7] Grammar-Aligned Decoding. Park et al.

---

### Meta-Review · Area_Chair_mXBf · 2026-01-06

**Summary:**

This paper introduces Constrained Adaptive Rejection Sampling (CARS), an extension of adaptive rejection sampling for constrained decoding. CARS can produce samples from the exact constrained distribution while eliminating invalid continuations. Via the evaluation of a variety of like program fuzzing and molecular generation, CARS consistently achieves higher efficiency.

**Reviewer Concerns:**

Concerns about technical novelty, experimental scope, and practical scalability are consistently raised.
- Reviewers JYYc and DrrA note that CARS appears to be a relatively straightforward extension of ARS, with limited differentiation from prior work such as IterGen (ICLR 2025) or MCMC-based approaches.
- The absence of comparison to ASAp (Park et al., 2024)—a relevant SOTA method mentioned only in passing—is a notable gap (Reviewer e4bz).
- Experiments are restricted to two mid-sized models (Qwen2.5-7B and Llama-3.1-8B), raising questions about generalizability (Reviewers JYYc and DrrA).

**Reviewer Scores:**

Reviewer DrrA may have a chance to raise the score.

---

### Decision · Program_Chairs · 2026-01-26

Reject